# Comparing Paclitaxel–Carboplatin with Paclitaxel–Cisplatin as the Front-Line Chemotherapy for Patients with FIGO IIIC Serous-Type Tubo-Ovarian Cancer

**DOI:** 10.3390/ijerph17072213

**Published:** 2020-03-26

**Authors:** Chen-Yu Huang, Min Cheng, Na-Rong Lee, Hsin-Yi Huang, Wen-Ling Lee, Wen-Hsun Chang, Peng-Hui Wang

**Affiliations:** 1Department of Obstetrics and Gynecology, Taipei Veterans General Hospital, Taipei 112, Taiwan; eu.huang501@gmail.com (C.-Y.H.); alchemist791025@gmail.com (M.C.);; 2Department of Obstetrics and Gynecology, National Yang-Ming University, Taipei 112, Taiwan; 3Institute of Clinical Medicine, National Yang-Ming University, Taipei 112, Taiwan; johnweiwang@gmail.com; 4Department of Nursing, Taipei Veterans General Hospital, Taipei 112, Taiwan; 5Biostatics Task Force, Taipei Veterans General Hospital, Taipei 112, Taiwan; sweethsin509@gmail.com; 6Department of Medicine, Cheng-Hsin General Hospital, Taipei 112, Taiwan; 7Department of Nursing, Oriental Institute of Technology, New Taipei City 220, Taiwan; 8Department of Medical Research, China Medical University Hospital, Taichung 440, Taiwan; 9Female Cancer Foundation, Taipei 104, Taiwan

**Keywords:** dose-dense weekly paclitaxel, epithelial tubo-ovarian cancer, FIGO stage IIIC, primary peritoneal serous carcinoma, triweekly carboplatin, triweekly cisplatin

## Abstract

The use of weekly chemotherapy for the treatment of patients with advanced-stage serous-type epithelial Tubo-ovarian cancer (ETOC), and primary peritoneal serous carcinoma (PPSC) is acceptable as the front-line postoperative chemotherapy after primary cytoreductive surgery (PCS). The main component of dose-dense chemotherapy is weekly paclitaxel (80 mg/m^2^), but it would be interesting to know what is the difference between combination of triweekly cisplatin (20 mg/m^2^) or triweekly carboplatin (carboplatin area under the curve 5-7 mg/mL per min [AUC 5-7]) in the dose-dense paclitaxel regimen. Therefore, we compared the outcomes of women with Gynecology and Obstetrics (FIGO) stage IIIC ETOC and PPSC treated with PCS and a subsequent combination of dose-dense weekly paclitaxel and triweekly cisplatin (paclitaxel–cisplatin) or triweekly carboplatin using AUC 5 (paclitaxel–carboplatin). Between January 2010 and December 2016, 40 women with International Federation of Gynecology and Obstetrics (FIGO) stage IIIC EOC, FTC, or PPSC were enrolled, including 18 treated with paclitaxel–cisplatin and the remaining 22 treated with paclitaxel–carboplatin. There were no statistically significant differences in disease characteristics of patients between two groups. Outcomes in paclitaxel–cisplatin group seemed to be little better than those in paclitaxel–carboplatin (median progression-free survival [PFS] 30 versus 25 months as well as median overall survival [OS] 58.5 versus 55.0 months); however, neither reached a statistically significant difference. In terms of adverse events (AEs), patients in paclitaxel–carboplatin group had more AEs, with a higher risk of neutropenia and grade 3/4 neutropenia, and the need for a longer period to complete the front-line chemotherapy, and the latter was associated with worse outcome for patients. We found that a period between the first-time chemotherapy to the last dose (6 cycles) of chemotherapy >21 weeks was associated with a worse prognosis in patients compared to that ≤21 weeks, with hazard ratio (HR) of 81.24 for PFS and 9.57 for OS. As predicted, suboptimal debulking surgery (>1 cm) also contributed to a worse outcome than optimal debulking surgery (≤1 cm) with HR of 14.38 for PFS and 11.83 for OS. Based on the aforementioned findings, both regimens were feasible and effective, but maximal efforts should be made to achieve optimal debulking surgery and following the on-schedule administration of dose-dense weekly paclitaxel plus triweekly platinum compounds. Randomized trials validating the findings are warranted.

## 1. Introduction

Epithelial tubo-ovarian cancer (ETOC) is the deadliest cancer among women, placing with fourth place of all the fetal diseases among women and ranking seventh of most common cancers in women’s cancer and women’s cancer-related deaths in Taiwan and globally in 2018 [1,2,3,4,5,6]. Serous-type is the most common subtype among EOCs in general; however, endometriosis-associated EOCs, such as clear cell type or endometrioid type are relatively common in certain populations, including Taiwan and Japan [7,8,9,10,11]. Clinically, the serous-type EOC and primary peritoneal serous carcinoma (PPSC) and primary Fallopian tube cancer (PFTC) are often considered the same group disease, based on the similar clinical behavior and possibly sharing the similar pathogenesis [12,13,14,15,16,17,18]. In 1996, McGuire and colleagues conducted a clinical trial to set up the standard therapy for patients with ETOC, including primary debulking surgery (PDS), also called primary cytoreductive surgery (PCS) plus adjuvant triweekly paclitaxel and cisplatin therapy [19]. The following studies have confirmed the superiority of survival benefits using paclitaxel into cisplatin-based regimens in place of cyclophosphamide for the treatment of patients with ETOC and PPSC [20,21]. Additionally, carboplatin in place of cisplatin in this combination has statistically significantly reduced the cisplatin-related potential toxicity of neural and renal systems and ameliorated the cisplatin-related high emetic effects [22,23,24,25]. 

### 1.1. Current Standard of Treatment

According to the National Comprehensive Cancer Network (NCCN) guideless version 1.2020 Ovarian Cancer/Fallopian Tube Cancer/Primary Perineal Cancer, principles of systemic therapy regimens for advanced-stage high-grade serous ETOC and PPSC included three parts [26]. Preferred regimens [26] include (1) the combination of intravenous (IV) paclitaxel 175 mg/m^2^ and IV carboplatin area under the curve ranging from 5 mg/mL per min (AUC 5) to 6 mg/mL per min (AUC 6) every three weeks (Q3W, also called triweekly) for 3-6 cycles [22,23,24,25]; (2) IV paclitaxel 175 mg/m^2^ day 1 followed by IV carboplatin AUC 6 Day 1 and bevacizumab 7.5 mg/kg Day 1 of cycle 1 Q3W for 5-6 cycles plus additional 12 cycles of bevacizumab Q3W for maintenance therapy; (3) IV paclitaxel 175 mg/m^2^ followed by IV carboplatin AUC 6 Day 1 Q3W for 6 cycles with 15 mg/kg bevacizumab added starting on Day 1 of cycle 2, Q3W for up to a total of 22 cycles [27,28,29,30,31]. 

Other recommended regimens [26] include the following four regimens, such as (1) IV paclitaxel 60 mg/m^2^ followed by IV carboplatin AUC 2 every week (weekly) [32,33], (2) IV docetaxel 60-75 mg/m^2^ and IV carboplatin AUC 5-6 Q3W [34,35], (3) IV carboplatin AUC 5 and IV pegylated liposomal doxorubicin 30 mg/m^2^ every four weeks (Q4W) [36,37,38], and (4) IV paclitaxel 80 mg/m^2^ weekly and IV carboplatin AUC 5 to AUC 7 triweekly [39,40,41,42]. 

The final part—which is especially useful in certain circumstances, such as in patients having optimally PCS stage II-IV diseases [26]—is the combination of IV paclitaxel 135 mg/m^2^ on Day 1, intraperitoneal (IP) cisplatin 75-100 mg/m^2^ on Day 2, and IV paclitaxel 60 mg/m^2^ on Day 8 Q3W for 6 cycles [43,44]. Although there are so many recommendations available as shown above, now, the most frequently acceptable standard chemotherapy for ETOC and PPSC is still the use of combination of triweekly carboplatin and paclitaxel [37,45,46,47,48,49,50,51]. 

However, the prognosis of patients with ETOC and PPSC is still disappointing, because of vague or free symptoms of patients, often misdiagnosed as less deadly gastro-enteral tract problems, and the absence of effective screening programs, and, in addition, its heterogeneous nature and different clinical development, accounting for late diagnosis in its advanced stages [52,53,54,55]. All result in a therapeutic challenge, contributing to low cure rate and high mortality rate, [1,2,3,4,5,19,20,21,22,23,24,25]. A median progression-free survival (PFS) is 16–21 months and an overall survival (OS) is 32–57 months [1,2,3,4,5,19,20,21,22,23,24,25]. Therefore, an improvement of therapeutic effect as well as a prolongation of PFS and/or OS is urgently needed. 

Based on the relative worse prognosis of advanced-stage ETOC and PPSC, many new modalities and strategies have been recently developed [56,57,58,59]. Although some of them have been already recommended by updated NCCN guidelines for ETOC and PPSC treatment [4,26], they are not widely used in the routine clinical practice [60]. Among these, induction chemotherapy (neoadjuvant chemotherapy- NACT) using either standard form of paclitaxel and carboplatin or its modification form, including dose-dense or high-dose chemotherapy, and in additional adding bevacizumab, has become more and more popular, especially for those patients who are not candidates for immediate PCS [61,62,63,64,65,66,67,68,69]. 

In addition, the changed delivery route or warm-up of chemotherapy agents (IP administration or IP hyperthermia therapy) has been also accepted in the certain clinical situations [43,44,70,71,72,73,74,75]. Moreover, advancing drug development, including poly(adenosine diphosphate (ADP)-ribose) polymerase inhibitors (PARPi) as well as small molecules targeting various kinds of signaling pathway has shown the dramatic improvement in ETOC and PPSC treatment [76,77,78,79,80,81,82,83,84,85,86,87]. Finally, immune checkpoint inhibitors or immune system modulators has also provided a chance for ETOC and PPSC patients, although the therapeutic effect is debated [88,89]. However, there is no doubt that the high cost of the aforementioned new therapeutic approaches will limit the clinical use in routine [90], and in addition, the compliance may be compromised by the requirement of maintenance or continuous long-term therapy in some drugs.

### 1.2. Gap in Knowledge of Current Standard of Therapy

Based on the negative impact on the high cost of many targeted therapies and the poor compliance of long-term maintenance therapy in certain population, as well as no differences or the improvement of clinical outcomes in additional drugs added to platinum/taxane combination or the use of different platinum doublets [91], dose-dense platinum/taxane might be a good alternative, due to the similar duration of completing therapy and acceptable cost expense. In addition, some evidence also favored this regimen as a feasible front-line chemotherapy for patients after PCS, based on the possible PFS benefits and considerably robust cost-effectiveness [92,93].

In theory, the maximal tolerated doses of chemotherapy may yield the highest cytotoxicity to tumors, with subsequent higher cure rates [94]. However, such treatment may need a longer treatment-free period to wait normal host cells for recovery [94]. Among these, hematopoietic progenitors might be the best example [94]. Without this treatment-free period, some catastrophic and life-threatened conditions might occur. During the holiday of drug, cancer cells and cells in the tumor microenvironment may also re-grow, possibly resultant development of aggressive behavior and chemo-resistance clone of tumor cells. All decided a rule of thumb to design most chemotherapy combinations to be repeated every three or four weeks [95]. The triweekly carboplatin plus paclitaxel regimen is just followed by the aforementioned rule, and this protocol is still one of the best-known and golden-standard regimens in the management of EOC patients after PCS. 

However, this “golden-standard” therapy only provides the relatively short PFS and unsatisfactory OS in these advanced-stage EOC patients, making the urgent needs of application of new strategies to improve therapeutic outcome and prolong the patients’ life. Since the use of 3-drug combinations and maintenance chemotherapy after the front-line chemotherapy is not recommended because of the absence of survival benefits but significantly increased adverse events (AEs) and possibly compromised quality of life [50,51,59], another strategy without the involving adding any new agents or targeted therapy is pursued [45]. Therefore, if single dose of treatment is decreased in their dosage, the holiday of drugs can be shortened. A dose-dense weekly prescription of drugs may fulfill the aforementioned requirement. 

### 1.3. Rationale of Dose-Dense Therapy

Dose-dense weekly chemotherapy is based on the concept of shortening the timing of recycling by Goldie-Coldman’s hypothesis and the Norton–Simon model [96,97,98], providing the rationale, such as a reducing interval between treatment, an increasing duration of chemotherapy exposure, and an increasing dose-intensity or dose-density of cytotoxic agents, and offering a chance that tumor regrowth could be decreased by this approach [99,100,101,102]. In addition, dose-dense chemotherapy can not only preserve the immune system but also promote the treatment-mediated tumor-specific immunity, especially the antitumor kill cluster of differentiation 8+ (CD8+) T-cell response [103]. Therefore, hematological markers have been evaluated to try to establish their predictive value [104,105,106,107]. Dose intensity strategies include increasing the dose per cycle, decreasing the cycle interval (dose densification) and decreasing the interval plus increasing the dose [108]. However, the latter two strategies involved shortening the gap between chemotherapy, which has proven decisive in some neoplasms such as breast cancer [99], and, of most importance, this more frequent administration of cytotoxic agents within short intervals has also decreased the chemotherapy-related death associated with the use of massive doses, such as infection or sepsis caused by severe neutropenia or spontaneous hemorrhage caused by severe thrombocytopenia [49].

Dr. Muggia proposed a very interesting hypothesis to show the potential benefits of dose-dense sequential treatment designs for first line, such as an initial dose-dense cisplatin and following dose-dense paclitaxel based on unpublished data from GOG-132 and ICON3 (Gynecologic Cancer InterGroup (GCIG) International Collaboration on Ovarian Neoplasms 3) in 2003 [109]. In the next year (2004), the same author found that increasing the dose of cisplatin above a certain threshold is not recommended in EOC patients because the greater toxicity with higher dose of cisplatin was found [110], although some studies did not support a significant increase toxicity after weekly dose-dense cisplatin treatment (median dose intensity with 32 or 45 mg/m^2^/week) [108]. However, the therapeutic effect of dose-dense platinum is still controversial. Some are in favor [111,112], but many are against the resultant benefits of dose-dense platinum [108,113,114,115,116]. Biweekly dose-dense carboplatin (AUC5) combined with paclitaxel (175 mg/m^2^) is not feasible based on dose-limiting toxicity, even though GCSF was used for support [116]. Previous studies that the main determinant of dose-dense treatment response was not achieved by the relative dose intensity of platinum itself [108]. Indeed, one report showed that the administration of dose up to cisplatin of 25 mg/m^2^/week might reach the plateau of dose-response curve [113]. 

By contrast, the toxicity of dose-dense taxane seemed to be well tolerated because of absence of relevant toxicity [94]. A pharmacokinetic study revealed that paclitaxel can be administered weekly at doses of 110 mg/m^2^ without interruption, while neutropenia precluded scheduled administration of doses ≥ 130 mg/m^2^ [94,117]. In theory, weekly dose-dense paclitaxel provided the better cytotoxicity to tumors, through more sustained exposure, limiting the emergence of tumors resistant to chemotherapy, enhancing the apoptotic and antiangiogenic effect, and improving the therapeutic index [94,118,119,120], although the single use of dose-dense paclitaxel for the first-line chemotherapy treatment of patients with ETOC seemed to be not approved by clinical trials [109,110].

Therefore, the application of a combination of taxane and platinum in ETOC has been conducted. In the management of all chemosensitive epithelial cancers, combination chemotherapy treatment has provided significant survival benefits compared to single agent chemotherapy when applied as initial therapy, and there is no doubt that the treatment of ETOC is similar in this regard [121]. Studies showed that combination of taxane and platinum compound appears both to attenuate the toxicity of the platinum compound and to facilitate the delivery of full dose on schedule and this full dose on schedule showed the strong correlation with patient outcomes [110]. Since both platinum and paclitaxel are the essential cytotoxic agents in ETOC patients after PCS, either any one or both can be administered as dose-dense protocol. Furthermore, dose-dense can be classified according to the types, including semiweekly dose-dense, in which paclitaxel was given weekly and carboplatin was given triweekly, and weekly dose dense, in which both paclitaxel and carboplatin were given weekly [47]. An accumulation dose of paclitaxel is 240–270 mg/m^2^ with separating into 80-90 mg/m^2^ per week compared to a single use of 175–180 mg/m^2^ every three weeks. By contrast, the dosage of carboplatin seems to be consistent with accumulation dose as AUC 5-7, regardless of administration triweekly (a single use of AUC 5-7) or weekly (AUC 2).

### 1.4. Previous Studies for Dose-Dense Therapy

Since the rational of dose-dense therapy supports the potential benefits for patients with advanced stage ETOC and PPSC, at least four large randomized clinical trials were conducted to test the hypothesis [32,33,39,40,41]. The Japanese Gynecologic Oncology Group (JGOG) 3016, named as JGOG 3016, provided a strong evidence to show the survival benefits in the combination regimen including dose-dense weekly paclitaxel (80 mg/m^2^) and triweekly carboplatin (AUC 6), which not only offered a longer PFS but also a significantly prolonged OS compared to those in a standard triweekly combination of paclitaxel and carboplatin treatment [32,33]. 

By contrast, this promising result was not reproducible in three other trials conducted in Western countries. The results of an NRG Oncology (the National Cancer Institute Cooperative Group Program plus the Radiation Therapy Oncology plus Gynecology Oncology Group)/GOG Study showed the similar PFS and OS in dose-dense regimen with adding 15 mg/kg bevacizumab treatment compared to the standard triweekly combination therapy [72]. MITO-7 (Multicentre Italian Trials in Ovarian cancer 7) showed that there was no statistically significant difference of median PFS between dose-dense weekly combination and standard triweekly combination therapy [32]. GOG-0262 showed no statistical difference of the median PFS between dose-dense weekly paclitaxel and triweekly carboplatin with/without adding 15 mg/kg bevacizumab treatment, respectively, and standard triweekly combination therapy [75]. The results from an ICON8 showed that the restricted mean PFS was not statistically significant different among the three groups (dose-dense or standard groups) [33]. 

A recent meta-analysis conducted by Marchetti and colleagues further updated these four randomized controlled trials containing 3698 patients, and found that dose-dense chemotherapy did not have a statistically significant benefit of PFS (HR 0.92, 95% CI 0.81-1.04) [47]. Additionally, the results were not changed with HR 1.01 (95% CI 0.93-1.10) and 0.82 (95% CI 0.63-1.08), respectively, even though the analysis was limited to both weekly and semi-weekly dose-dense data [47]. Therefore, the authors believed that conventional triweekly combination chemotherapy is still one of the golden-standard treatments of patients with advanced stage ETOC [47].

Why are the survival benefits only apparent in certain populations, such as Japanese or others? Although the real reasons associated with conflicted results are uncertain, genomic background of race, tumor heterogeneity, administration schedule, cisplatin or carboplatin, and the duration to complete postoperative adjuvant chemotherapy may all be attributable. Among these, we believe that platinum compound may play a certain role in the discrepancy. 

In order to evaluate the effects and safety of dose-dense of paclitaxel combining with either cisplatin or carboplatin, the following retrospective study was conducted. Patients with FIGO IIIC serous-type ETOC and PPSC treated either with triweekly cisplatin (20 mg/m^2^) plus dose-dense weekly paclitaxel (80 mg/m^2^) or triweekly carboplatin (AUC 5) plus dose-dense weekly paclitaxel (80 mg/m^2^) as postoperative adjuvant chemotherapy after PCS were retrospectively reviewed. However, nearly all dose-dense paclitaxel-platinum combination chemotherapy regimens used as the front-line postoperative adjuvant therapy are focused on the dose-dense paclitaxel, without consideration of either the platinum compounds or the dose-dense agents.

## 2. Materials and Methods

### 2.1. Patient Population

After institutional review board approval (VGHIRB 2019-07-039BC), all patients between January 2010 and December 2016 who fulfilled the following inclusion (International Federation of Gynecology and Obstetrics (FIGO) stage IIIC histologically confirmed high-grade serous-type ETOC and PPSC, an initial PCS, a total of six cycles of weekly paclitaxel [80 mg/m^2^] plus either triweekly cisplatin [20 mg/m^2^] or triweekly carboplatin [AUC 5] regimen) and exclusion (NACT, other newly diagnosed cancer, previous chemotherapy, or radiotherapy in the past two years; incomplete chemotherapy or the delayed of the first course chemotherapy (>7 days after PCS), simultaneous use of other antineoplastic agents, antiangiogenic agents, or targeted therapy) criteria were identified from our gynecologic oncology registry. Since both cisplatin and carboplatin are available in our department, the selection of cisplatin or carboplatin was based on the patient’s will after shared decision making.

### 2.2. Treatment

All patients underwent PCS initially and then were treated with weekly paclitaxel (80 mg/m^2^) plus either triweekly cisplatin (20 mg/m^2^) or triweekly carboplatin (AUC 5). Paclitaxel was administered over 2 h intravenously on days 1, 8, and 15, and either cisplatin for 1 h or carboplatin for 1 h on day 1 was infused intravenously. Premedication including dexamethasone (20 mg), 2000 ml of normal saline, and palonosetron (250 *u*g) was prescribed intravenously during the chemotherapy therapy day. When grade 3/4 neutropenia occurred, the patients were treated with granulocyte colony-stimulating factor (GCSF) for three days before chemotherapy. However, treatment was delayed if febrile neutropenia or grade 3/4 neutropenia without correction was noted. If a three-fold increase in liver enzymes of patients was detected, paclitaxel (60 mg/m^2^) was used. The cisplatin dose was decreased to half dosage if the estimated glomerular filtration rate (eGFR) was decreased to 45–60 ml/min, which was calculated using the Cockcroft–Gault formula [122,123,124]. 

### 2.3. Assessments

The first course chemotherapy was prescribed within one week after PCS. The clinical symptoms or signs—including the National Cancer Institute’s Common Terminology Criteria for Adverse Events (NCI-CTCAE), version 5.0Adverse effects (AEs)—were used to evaluate the severity of adverse events before every cycle of treatment [125,126]. During the follow-up, the patients received both image evaluation, such as ultrasound, computed tomography, or magnetic resonance imaging and biochemical markers, such as CA-125 (the abbreviation of cancer antigen 125, carcinoma antigen 125, or carbohydrate antigen 125) examinations. The evaluation of image finding was based on Response Evaluation Criteria in Solid Tumors (RECIST), version 1.1 [127,128,129]. Criteria of progression or recurrence were based on findings of clinical symptoms or signs accompanied with image findings and tumor markers described elsewhere [41,130,131]. 

### 2.4. Statistical Analysis

The primary endpoint was PFS, defined as the time from the date patients first underwent PCS to the earliest date of disease progression, death from any cause, or the date of the last known follow-up. The secondary endpoint was OS, defined as the time from the date patients first underwent PCS to the date of death from any cause or the date of the last known follow-up. The Kaplan–Meier method was used to generate survival curves, and the log-rank test was used to detect the differences between survival curves. Prognostic factors for PFS or OS were evaluated using Cox proportional hazard methods. Multivariate analysis using Cox stepwise forward regression was conducted for the covariates selected in univariate analysis. A *p* value <0.05 was considered to be statistically significant. All statistical analyses were conducted with SAS version 9.3 (SAS Institute, Cary, NC) and Stata Statistical Software, version 12.0 (Stata Corporation, College Station, TX).

## 3. Results

### 3.1. Clinical Characteristics and Pathological Status

A total of 40 women with International Federation of Gynecology and Obstetrics (FIGO) stage IIIC ETOC or PPSC were analyzed, including 18 treated with paclitaxel–cisplatin and the remaining 22 treated with paclitaxel–carboplatin. Table 1 summarizes the characteristics of patients in each group. The mean age of the whole population was 59 years. Optimal PCS was achieved in 55% in overall, and 54.5% and 55.6% in the paclitaxel–carboplatin and paclitaxel–cisplatin groups, respectively. Patients in the paclitaxel–carboplatin group had a higher risk of a prolonged time to finish 6 cycles of the front-line chemotherapy after PCS than those in paclitaxel–cisplatin group (45.5% versus 11.1%, p = 0.018), which reached the statistically significant difference.

### 3.2. Adverse Events (AEs)

Adverse events (AEs) are listed in Table 2. In the current study, no treatment-related death was found. The most common all-grade AEs in the entire cohort included anemia, nausea, neutropenia, and peripheral neuropathy. However, the occurrence of each AE was different in both groups. Patients who were treated with the paclitaxel–carboplatin regimen had a higher risk of development of neutropenia than those with the paclitaxel–cisplatin regimen. In addition, grade 3/4 neutropenia occurred more frequently in patients treated with the paclitaxel–carboplatin regimen compared to that in patients with the paclitaxel–cisplatin regimen. Both reached the statistically significant difference. In current study, cisplatin-related AEs, such as renal toxicity, neurotoxicity, nausea, or vomiting were mild or absent, as shown in Table 2.

### 3.3. Outcomes

During the whole study period with a median follow-up time of 55 months (at the time of the data cutoff on 30 June 2019), disease progression and/or disease-related death occurred in 23 patients (57.5%). As was presented in Figure 1, the median PFS was 25.0 months in patients treated with the paclitaxel–carboplatin regimen, and 30.0 months in patients treated with the paclitaxel–cisplatin regimen, without a statistically significant difference. The median OS was 55.0 months and 58.5 months in patients treated with paclitaxel–carboplatin and paclitaxel–cisplatin regimens, respectively (Figure 2). There was also no statistically significant difference between the two groups. 

### 3.4. Prognostic Factors

To identify the prognostic factors for disease progression (PFS), a univariate analysis of clinic-pathologic factors showed that residual tumors more than 1 cm in size and the prolonged completion of postoperative adjuvant chemotherapy were associated with worse prognosis (Table 3). Both these factors were also independent risk factors when using the multivariate analysis model.

Further evaluation was performed to analyze the prognostic factors associated with overall survival and the results showed that optimal debulking surgery was the most important independent prognostic factor to predict better outcomes in patients with FIGO IIIC ETOC and PPSC. However, the on-schedule administration of postoperative adjuvant chemotherapy regardless, in-time and on-time chemotherapy was also a very important predictor in those patients who had a better chance for prolonged OS (Table 4). 

## 4. Discussion

### 4.1. Main Findings

The main findings of the current study showed that both regimens could be successfully used for the treatment of patients with FIGO IIIC ETOC or PPSC, offering 30 and 25 months of median PFS in the paclitaxel–cisplatin and paclitaxel–carboplatin groups, respectively. The median OS was 58.5 months in the paclitaxel–cisplatin group and 55.0 months in the paclitaxel–carboplatin group. Although neither PFS nor OS reached the statistically significant difference, it seemed to be little better in patients treated with the paclitaxel–cisplatin regimen compared to those treated with the paclitaxel–carboplatin regimen. However, to compare the frequency and severity of AEs, we found that patients in the paclitaxel–carboplatin group had more AEs than those in paclitaxel–cisplatin group did, including a higher risk of neutropenia and grade 3/4 neutropenia, and the need for a longer period to complete the front-line chemotherapy. The need for a longer period to finish the dose-dense therapy seemed to be associated with a worse outcome for patients. Further analysis showed that the period between the first-time chemotherapy to the last dose (6 cycles) of chemotherapy > 21 weeks was associated with worse prognosis in patients compared to that ≤21 weeks, with hazard ratio (HR) of 81.24 for PFS and 9.57 for OS. As predicted, suboptimal debulking surgery (>1 cm) also contributed to the worse outcome than optimal debulking surgery (≤1 cm) with a HR of 14.38 for PFS and 11.83 for OS (Table 3 and Table 4).

### 4.2. Summary of Studies Addressing Dose-Dense therapy: Survival Outcome (Table 5)

Some early phase studies have also been conducted to test the tolerability and efficacy of dose-dense weekly paclitaxel (80 mg/m^2^) and triweekly carboplatin (AUC 5) with other agents [132,133]. For example, one phase II study modified a dose-dense paclitaxel (80 mg/m^2^) and carboplatin (AUC 5) every four weeks for the treatment of advanced-stage ETOC patients, and found that the median PFS and OS were 22.5 and 31.5 months, respectively [132]. Another phase II study adding bevacizumab in dose-dense chemotherapy (weekly paclitaxel 80 mg/m^2^ and triweekly carboplatin AUC 5) showed that the median PFS was 16.9–22.4 months in advanced-stage ETOC patients [133]. 

Based on the impressive survival benefits of the earlier reports, the prospective, randomized trials may be the best method to test the efficacy and safety of dose-dense chemotherapy in the management of ETOC patients. The first large study, named as the JGOG 3016 has shown the survival benefits in dose-dense regimen containing weekly paclitaxel (80 mg/m^2^) and triweekly carboplatin (AUC 6), which not only offered a longer PFS (median PFS of 28.2 months, 95% CI 22.3-33.8 months) but also a significantly prolonged OS (median OS of 100.5 months, 95% CI 65.2-∞ months) than those in the standard triweekly paclitaxel–carboplatin regimen [32,33]. 

The results of an NRG Oncology/GOG Study showed a similar PFS (the median PFS of 24.9 months) and OS (the median OS of 75.5 months) in dose-dense weekly paclitaxel (80 mg/m^2^) and triweekly carboplatin (AUC 6) with 15 mg/kg bevacizumab treatment compared to the standard triweekly combination therapy [72]. 

MITO-7 has shown that there was no statistically significant difference in median PFS between dose-dense weekly combination (paclitaxel 60 mg/m^2^ plus carboplatin AUC 2) and standard triweekly combination therapy (18.3 months [95% CI 16.8-20.9 months] versus 17.3 months [95% CI 15.2-20.2 months]) with a HR of 0.96, 95% CI 0.80-1.16 [32]. 

GOG-0262 showed the median PFS of 14.9 and 14.2 months in dose-dense weekly paclitaxel (80 mg/m^2^) and triweekly carboplatin (AUC 6) with/without adding 15 mg/kg bevacizumab treatment, respectively, without a statistically significant difference compared to the standard triweekly combination therapy [41]. 

The results from an ICON8 have shown that the median PFS was 20.8 months (95% CI 11.9-59.0 months) and 21.0 months (95% CI 12.0-54.0 months) in dose-dense weekly paclitaxel (80 mg/m^2^) and triweekly carboplatin (AUC 5 or 6) and in dose-dense weekly paclitaxel (80 mg/m^2^) and weekly carboplatin (AUC 2) regimen, respectively, and none were statistically significant different from the 17.7-month median PFS in the standard-dose triweekly paclitaxel and triweekly carboplatin (AUC) treatment [33]. 

Although the evidence would be much more powerful if the results were obtained from prospective, randomized trials, the management of the individual patient who does not fulfill the criteria of a clinical trial or is enrolled into the clinical trial will be debated continuously. In addition, the strict criteria of a clinical trial, such as inclusive and exclusive criteria, may not reflect the real situations of patients. Therefore, clinical practices are not uniform, and retrospective evaluations may partly reveal real-world clinical practice. For example, the elderly population is often excluded in clinical trials. Dr. Bun and colleagues found dose-dense weekly paclitaxel plus triweekly carboplatin was feasible for elderly patients, although severe neuropathy might occur more frequently than in younger groups [134]. In fact, the elderly population might have a higher risk of receiving nonstandard chemotherapy [135].

### 4.3. Dose-Dense Therapy-Related Adverse Events: Prefer the Use of Low-Dose Cisplatin in Place of Carboplatin in Platinum-based Therapy

It is believed that the use of carboplatin in place of original cisplatin in the platinum compound-based paclitaxel doublet might provide a better quality of life (QOL) in the long-term, since this regimen—with carboplatin dosed using the Calvert formula—yielded convincing non inferior outcomes when compared with the prior, more toxic, cisplatin–paclitaxel regimen [136]. By contrast, bone marrow suppression, including the toxicity of hematological system might be more apparent in carboplatin administration, although Boyd and Muggia proposed that carboplatin–paclitaxel therapy is generally safe when the drug is properly dosed [136]. The occurrence of neutropenia is one of the most common causes necessitating postponing the therapy or modifying (decreasing) the dosage of therapeutic agent. One study from Japan in 2006 tried to determine the feasibility of docetaxel–cisplatin therapy Q3W compared with docetaxel–carboplatin therapy Q3W as a front-line for ETOC patients [136]. The results, as predicted, showed that the incidence of grade 4 neutropenia was much higher in the docetaxel–carboplatin group than that in the docetaxel–cisplatin group (74% versus 39%), suggesting the feasibility of docetaxel–cisplatin combination therapy as front-line therapy for ETOC patients [136]. In addition, the rationale of the use of “dose-dense” in place of “standard-dose” is the attempt to decrease the incidence of chemotherapy-related severe hematotoxicity, such as infection or sepsis caused by severe neutropenia or spontaneous hemorrhage caused by severe thrombocytopenia [49]. In Taiwan, GCSF is not routinely prescribed and insurance does not cover the cost if GCSF is used in a prophylactic role. Furthermore, the obstacle is still present when facing patients with chemotherapy-related thrombocytopenia [137,138,139]. 

Thrombocytopenia is a principal consideration, as well as the dose-limiting toxicity of carboplatin [140,141]. Although thrombocytopenia unlikely occurs in chemotherapy-naïve patients, this toxicity usually begins to appear after day 14 and is predictably cumulative, contributing to the consideration of a carboplatin dose reduction [140]. Due to aforementioned limitation, some gynecological oncologists favored the use of cisplatin as the drug in the platinum-based combination therapy, except for those patients with identified impaired renal function (eGFR < 60 ml/min). The current low-dose cisplatin plus dose-dense paclitaxel regimen was associated with lower rates of hematologic toxicities compared to the conventional dose-dense paclitaxel plus carboplatin regimen. The results from an ICON8 have also shown that patients treated with a weekly regimen had increased grade 3 or 4 toxic effects, although these high-grade toxicities were predominantly uncomplicated [33]. However, both regimens are carboplatin-based paclitaxel combinations. 

Consistent with the increased toxicity of the use of carboplatin in previous studies [136], the incidence of grade 3/4 neutropenia was significantly lower in the cisplatin–paclitaxel arm than in that of the carboplatin–paclitaxel arm (27.8% versus 77.3%, p = 0.002) in our study. In addition, other parameters of hematological examination, including grade 3/4 anemia and grade 3/4 thrombocytopenia also favored the cisplatin–paclitaxel regimen with incidence of 5.6% and none, respectively, compared to 22.7% and 13.6% in the carboplatin–paclitaxel regimen in the current study. We believe that this carboplatin–paclitaxel combination therapy-related hematotoxicity might explain why more patients in the carboplatin–paclitaxel arm had a prolongation of the period between the initial first dose of chemotherapy and the final dose of chemotherapy (six cycles of chemotherapy) than those in the cisplatin–paclitaxel arm (45% versus 11.1%, p = 0.018). Further survival analysis showed that this prolongation of the therapeutic period might be associated with a worse outcome in FIGO IIIC ETOC and PPSC patients.

By contrast, the cisplatin-related renal, neural, and gastrointestinal system toxicity was lower in the current study. The main reason is the use of low-dose cisplatin (20 mg/m^2^) as a platinum compound in the platinum–paclitaxel combination therapy.

### 4.4. The Benefits of Maximal Cytoreductive Surgery and The Consideration of the Location of Residual Tumors

In agreement with well-known concepts [142,143,144,145,146], a rapid reduction in the tumor burden is the guarantee of successful treatment in cancer patients. Nearly all studies have confirmed that maximal cytoreduction surgery is the most critical step in the management of patients with advanced-stage ETOC or PPSC [142,143,144,145,146]. According to the definition of the Gynecologic Oncology group (GOG) for ’optimal’ as having residual tumor nodules which should be ≤ 1 cm in size, the Cochrane review revealed that the outcome of ETOC patients with residual disease ≤ 1 cm is better than of those with residual disease > 1 cm [143], although it is well known that PCS to no gross residual tumor might be associated with the longest PFS and OS [144,145]. One study showed an interesting finding that patients still had diverse outcomes despite undergoing optimal PCS (≤ 1 cm residual disease), in which it demonstrated that patients with a tumor limited in a single anatomic location had better median PFS and OS than patients with tumors widely distributed in multiple anatomic locations [146]. 

In our current study, we also tested the aforementioned findings. We found that patients with suboptimal PCS with a residual tumor > 1 cm indeed had a worse prognosis, with a hazard ratio (HR) of 14.38 (95% confidence interval [CI] 4.18-49.46) in PFS and a HR of 11.83 (95% CI 1.48-94.72) in OS, compared with those patients who could achieve optimal PCS with a residual tumor of a size less than 1 cm, supporting the rationale that small-volume tumors are more prone to clearance [61]. However, the widespread nature of tumors seemed to not be an independent risk factor for worse prognosis. We found that there was no statistically significant difference of outcome in patients whose residual tumors were in localized anatomical site or in multiple anatomical sites.

### 4.5. The Strengths and Limitations

The current study has the following strengths: All patients were serous-type ETOC and PPSC with FIGO stage IIIC. The follow-up period was long enough, of nearly 5 years (a mean follow-up period of 55 months). All patients were treated with by one senior doctor (P.-H.W). All the suggested study population was homogeneous. 

There are some limitations in the current study. Retrospective data collection in nature may not be strong enough to show clear evidence. However, as shown above, this is a real-world clinical practice. In addition, only patients who underwent PCS with surgical confirmation of FIGO IIIC serous-type ETOC and PPSC and completed 6 cycles of dose-dense paclitaxel-platinum compounds were included, contributing to the risk of selection bias. Furthermore, the number of patients in the current study was small in each arm, contributing to wide confidence intervals. It demonstrated the higher risk of low precision in the estimates. Therefore, the findings of the current study should be used with caution. Finally, new therapeutic strategies—such as maintenance therapy, NACT, or intraperitoneal therapy—were not evaluated in the current report. However, we believe none of them impede the value of the current study. 

## 5. Conclusions

Our study found that paclitaxel might be one of the most critical components for the successful administration of dose-dense adjuvant chemotherapy. In addition, the on-schedule delivery and shortening of the entire therapeutic period are a critical step for the survival benefits of patients who undergo a postoperative adjuvant dose-dense paclitaxel plus platinum compounds regimen. Of course, maximal efforts should be made to eradicate the tumor as much as possible. Although recent evidences highlighted the value of precise medicine, personalized and individual therapy, along with targeted therapy for cancer patients [147,148,149,150,151]—the net health benefits (NHBs), cost-effectiveness or quality of life may reflect the relative value of treatment options in EOC [100,152,153,154,155,156,157]. With the low risk of AEs, shortened therapeutic period, and, of most importance, working without compromising therapeutic effects—low-dose triweekly cisplatin plus dose-dense weekly paclitaxel might be associated with better NHBs, a better cost-effectiveness, and a better QOL compared to a carboplatin-based dose-dense paclitaxel regimen. More studies are encouraged to test our findings.

## Figures and Tables

**Figure 1 ijerph-17-02213-f001:**
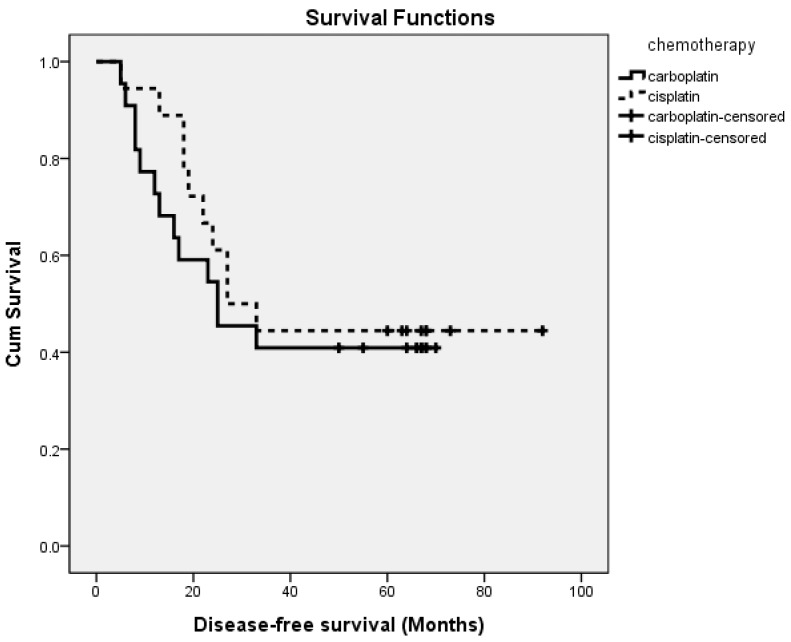
Progression-free survival curves for patients treated either with carboplatin-based dose-dense chemotherapy or with cisplatin-based dose-dense chemotherapy.

**Figure 2 ijerph-17-02213-f002:**
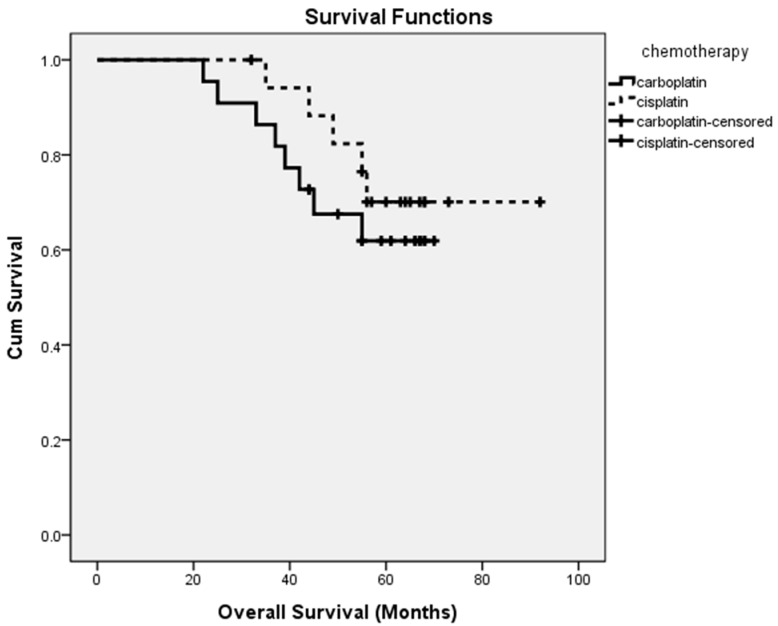
Overall survival curves of patients for patients treated either with carboplatin-based dose-dense chemotherapy or with cisplatin-based dose-dense chemotherapy.

**Table 1 ijerph-17-02213-t001:** Characteristics of the patients with FIGO IIIC serous type epithelial tubo-ovarian cancer, or primary peritoneal carcinoma treated with weekly paclitaxel (80 mg/m^2^) plus either carboplatin (AUC 5) or cisplatin (20 mg/m^2^) combination chemotherapy triweekly.

	Carboplatin	Cisplatin	*p*
Number of patients	22	18	
Age (years)	58.5 ± 9.4	59.4 ± 9.4	0.768
Size of residual tumors			0.949
≤1cm	12 (54.5%)	10 (55.6%)	
>1cm	10 (45.5%)	8 (44.4%)	
Site of residual tumor			0.676
Localized	12 (54.5%)	11 (61.1%)	
Whole abdominal cavity	10 (45.5%)	7 (38.9%)	
Period to complete the front-line chemotherapy			0.018
≤21 weeks	12 (54.5%)	16 (88.9%)	
>21 weeks	10 (45.5%)	2 (11.1%)	
ECOG			0.884
0-1	21 (95.5%)	17 (94.4%)	
2-3	1 (4.5%)	1 (5.6%)	

Carboplatin: carboplatin (AUC 5)-based dose dense chemotherapy; Cisplatin: cisplatin (20mg/m^2^)-based dose dense chemotherapy; ECOG: Eastern Cooperative Oncology Group Performance Status. Data are presented as a number (%) or the mean ± standard deviation.

**Table 2 ijerph-17-02213-t002:** Adverse events the patients with FIGO IIIC serous type epithelial tubo-ovarian cancer, or primary peritoneal carcinoma treated with weekly paclitaxel (80 mg/m^2^) plus either carboplatin (AUC 5) or cisplatin (20 mg/m^2^) combination chemotherapy triweekly.

Events	Any grade, n (%)			Grade 3/4, n (%)		
	CARBO	CIS	*p*	CARBO	CIS	*p*
Neutropenia	20 (90.9)	7 (38.9)	< 0.0001	17 (77.3)	5 (27.8)	0.002
Anemia	21 (95.5)	18 (100)	0.360	5 (22.7)	1 (5.6)	0.130
Thrombocytopenia	5 (22.7)	3 (16.7)	0.634	3 (13.6)	0	0.103
Renal toxicity	1 (4.5)	3 (16.7)	0.204	0	0	
Proteinuria	5 (22.7)	1 (5.6)	0.130	0	0	
Peripheral neuropathy	8 (36.4)	7 (38.9)	0.870	0	0	
Nausea	9 (40.9)	9 (50.0)	0.565	0	0	

CARBO: carboplatin (AUC 5)-based dose dense chemotherapy; CIS: cisplatin (20mg/m^2^)-based dose dense chemotherapy; n: number of patients; data are presented as numbers and percentages.

**Table 3 ijerph-17-02213-t003:** Association between baseline characteristics and progression-free survival.

Parameters		Univariate Analysis		Multivariate Analysis	
	*n*	HR (95% CI)	*p*	HR (95% CI)	*p*
**Size**					
≤1cm	22	1(Ref)		1(Ref)	
>1cm	18	8.68 (3.23–23.35)	<0.0001	14.38 (4.18–49.46)	<0.0001
**Site**					
Localized	23	1 (Ref)		1 (Ref)	
WAC	17	1.38 (0.61–3.14)	0.440	0.71 (0.28–1.82)	0.470
**Period**					
≤21 weeks	28	1 (Ref)		1 (Ref)	
>21 weeks	12	28.49 (8.36–97.06)	<0.0001	81.24 (14.03–470.31)	<0.0001

n: number of patients; HR: hazard ratio; 95% CI: 95% confidence interval; Size: size of residual tumor; Site: site of residual tumor; Localized: a single anatomic site; WAC whole abdominal cavity similar to multiple anatomic sites; Period: period between the initial first time of chemotherapy and the end of final chemotherapy (six cycles of chemotherapy); Ref: reference.

**Table 4 ijerph-17-02213-t004:** Association between baseline characteristics and overall survival.

Parameters		Univariate Analysis		Multivariate Analysis	
	*n*	HR (95% CI)	*p*	HR (95% CI)	*p*
**Size**					
≤1cm	22	1(Ref)		1(Ref)	
>1cm	18	22.37 (2.88–173.86)	0.003	11.83 (1.48–94.72)	0.020
**Site**					
Localized	23	1 (Ref)		1 (Ref)	
WAC	17	2.41 (0.79–7.39)	0.122	2.54 (0.76–8.53)	0.131
**Period**					
≤21 weeks	28	1 (Ref)		1 (Ref)	
>21 weeks	12	15.31 (4.13–56.78)	<0.0001	9.57 (2.34–39.18)	0.002

n: number of patients; HR: hazard ratio; 95% CI: 95% confidence interval; Size: size of residual tumor; Site: site of residual tumor; Localized: a single anatomic site; WAC whole abdominal cavity similar to multiple anatomic sites; Period: period between the initial first time of chemotherapy and the end of final chemotherapy (six cycles of chemotherapy); Ref: reference.

**Table 5 ijerph-17-02213-t005:** Summary of treatment efficacy and safety of dose-dense weekly paclitaxel plus platinum compounds in the management of patients with epithelial tubo-ovarian cancer and primary peritoneal serous carcinoma.

Authors	Stage	*n*	Regimen	PFS	OS
Prospective randomized trials					
Katsumata [40]	II–IV	312	P 80 mg/m^2^ (D1,8,15), C 6 (D1)	28.2 (M)	100.5 (M)
Pignata [32]	IC-IV	406	P 60 mg/m^2^ (D1,8,15), C 2 (D1,8,15)	18.3 (M)	
Chan [41]	II–IV	340	P 80 mg/m^2^ (D1,8,15), C 6 ±BEV (D1)	14.7 (M)	-
		55	P 80mg/m^2^ (D1,8,15), C 6 (D1)	14.2 (M)	-
Clamp [33]	I-IV		P 80 mg/m^2^ (D1,8,15), C 5,6 ± BEV (D1)	20.8 (M)	
			P 80 mg/m^2^ (D1,8,15), C 2 (D1,8,15) ± BEV (D1)	21.0 (M)	
Walker [72]	II–IV	521	P 80 mg/m^2^ (D1,8,15), C 6 +BEV (D1)	24.9 (M)	75.5 (M)
**Retrospective study, including phase II study**					
Abaid [132]	III-IV	88	P 80 mg/m^2^ (D1,8,15), C 5 (D1), stop one week	22.5 (M)	31.5 (M)
Fleming [133]	III-IV	33	P 80 mg/m^2^ (D1,8,15), C 5 + BEV (D1)	16.9-22.4 (M)	
Murphy [102]	III	38	P 80 mg/m^2^ (D1,8,15), C 5 (D1)	31.3 (m)	54.5 (m)
Boraska Jelavić [105]	I-IV	43	P 80 mg/m^2^ (D1,8,15), C 5 (D1)	20-24 (M)	
Rettenmaier [78]	I-IV	100	P 80 mg/m^2^ (D1,8,15), C 5 (D1)	27.6 (M)	
Cheng [101]	IIIC-IV	32	P 80 mg/m^2^ (D1,8,15), Cisplatin 20 mg/m^2^ (D1)	27.0 (M)	56 (m)
Current study	IIIC	18	P 80 mg/m^2^ (D1,8,15), Cisplatin 20 mg/m^2^ (D1)	30.0 (M)	58.5 (M)
		22	P 80 mg/m^2^ (D1,8,15), C 5 (D1)	25.0 (M)	55.0 (M)

Dose-dense chemotherapy is limited on the intravenous route. RCT: randomized control trial, containing co-administration of bevacizumab (BEV) 15 mg/kilograms at day 1 triweekly and patients treated with neoadjuvant chemotherapy; *n*: number of patients; PFS: median (M) or mean (m) progression-free survival calculated by months; OS: median or mean overall survival calculated by months; Stage: FIGO stage (International Federation of Gynecology and Obstetrics stage); P: paclitaxel; D: day; C 2, C 5 or C 6: carboplatin (AUC 2, AUC 5 or AUC 6: area under the curve 2, 5, or 6 mg/mL per minute).

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
