# Peer review of "Comparing Paclitaxel–Carboplatin with Paclitaxel–Cisplatin as the Front-Line Chemotherapy for Patients with FIGO IIIC Serous-Type Tubo-Ovarian Cancer"

_ijerph, 2020, doi:10.3390/ijerph17072213_

Round 1

Reviewer 1 Report

The work is interesting and well written. The research project and the experimental design are appropriate, the conceptional structure is well organized. In the present form, the manuscript can be accepted with minor changes because there are some aspects needing to clarify.

Point by point

Introduction

Line 84. I don’t understand the references in brackets: 30-104? Please, clarify this enumeration. 

Discussion

In my opinion this section seems to be too speculative and repetitive in some aspects. You could summarize some concepts.

Author Response

Manuscript ID IJERPH-731096R1 and title: Compare Dose-Dense Paclitaxel Plus either Standard-Dose Carboplatin or Low-Dose Cisplatin as the Front-Line Regimen in the Management of FIGO IIIC Serous-Type Ovarian, Fallopian Tube, and Primary Peritoneal Cancer

Peng-Hui Peter Wang, M.D., Ph.D.,

Professor and Director

Department of Obstetrics and Gynecology,

Taipei Veterans General Hospital, and National Yang-Ming University School of Medicine

201, Shih-Pai Road, Section 2, Taipei 112, TAIWAN

Telephone: 886-2-2875-7566; Fax: 886-2-5570 2788

E-mail: phwang@vghtpe.gov.tw; pongpongwang@gmail.com

Special Issue: Gynecological Cancer

Date: 20 March 2020

Dear Editors and Reviewer I

Thank you very much for your information by email on date of 18 March 2020. We are very happy to have a chance to revise our article with manuscript number of IJERPH-731096R1 and a title as “Compare Dose-Dense Paclitaxel Plus either Standard-Dose Carboplatin or Low-Dose Cisplatin as the Front-Line Regimen in the Management of FIGO IIIC Serous-Type Ovarian, Fallopian Tube, and Primary Peritoneal Cancer, which is submitted to the Special Issues “Gynecological Cancer” of the International Journal of Environmental Research and Public Health (ISSN 1660-4601) for your consideration.

We followed the suggestions made by experts (reviewers and academic editor) International Journal of Environmental Research and Public Health Editorial Office to revise this manuscript, which includes an itemized, point-by-point response to each reviewer, including the reviewer's original comment(s). We specify the changes made to address each of their concerns and we also include the changes made in the response and indicate the locations in the manuscript. We address each reviewer’s comment using the reviewer's number. Please see the following pages at the end of this letter.

We also agree to pay to have our article published under the Open Access option if the article fulfills the criteria for publication.

All co-authors have approved this manuscript and agreed to submit this original article to the International Journal of Environmental Research and Public Health. We also agree to pay to have our article published under the Open Access option if the article fulfills the criteria for publication. That is to say that we agree that the article processing charges (APC) of Swiss Francs apply to accepted papers. We totally agree the following: this is an open access article distributed under the Creative Commons Attribution License 4.0, which permits unrestricted use, distribution, and reproduction in any medium, provided the original work is properly cited.

All authors certify that this manuscript is a unique submission and is not being considered for publication by any other source in any medium. Further, the manuscript has not been published, in part or in full, in any form.

We appreciate your kindness for further considering it for publication. We are looking forward to hearing from you soon.

Very sincerely yours,

Peng-Hui Wang, MD, PhD

Chen-Yu Huang, (eu.huang501@gmail.com); Min Cheng, (alchemist791025@gmail.com); Na-Rong Lee, (nllee@vghtpe.gov.tw); Hsin-Yi Huang (sweethsin509@gmail.com); Wen-Ling Lee, (johnweiwang@gmail.com); Wen-Hsun Chang (whchang818@gmail.com)

Response to reviewer I’ comments

Open Review

English language and style

( ) Extensive editing of English language and style required  
( ) Moderate English changes required  
(x) English language and style are fine/minor spell check required  
( ) I don't feel qualified to judge about the English language and style 

Response: Thank you very much for your favorable consideration, and text has been already carefully edited.

Yes

Can be improved

Must be improved

Not applicable

Does the introduction provide sufficient background and include all relevant references?

X

Is the research design appropriate?

X

Are the methods adequately described?

X

Are the results clearly presented?

X

Are the conclusions supported by the results?

X

Response: Thank you very much for your favorable consideration. We have followed your valuable comments to re-organize the current study and the response to every comment is shown below point-by-point. Thank you again. Please read:

Comments and Suggestions for Authors

The work is interesting and well written. The research project and the experimental design are appropriate, the conceptional structure is well organized. In the present form, the manuscript can be accepted with minor changes because there are some aspects needing to clarify.

Response: It is so nice to hear it. As shown above, we have followed your valuable comments to re-organize the current study and describe our presentation clearly.

Point by point

Introduction

Line 84. I don’t understand the references in brackets: 30-104? Please, clarify this enumeration. 

Response: Thank you very much and we appreciate your valuable comments. Although these cited references are important, we totally agree with your recommendation. We cited every item one by one and deleted some references to make it clear and informative. Please read:

1. Introduction

Epithelial tubo-ovarian cancer (ETOC) is the deadliest cancer among women placing with 4th place of all the fetal diseases among women and ranking 7th of most common cancer in women cancer and women’s cancer-related death in Taiwan and globally in 2018 [1-6]. Serous-type is the most common subtype among EOCs in general; however, endometriosis-associated EOCs, such as clear cell type or endometrioid type are relatively common in certain populations, including Taiwan and Japan [7-11]. Clinically, the serous-type EOC and primary peritoneal serous carcinoma (PPSC) and primary Fallopian tube cancer (PFTC) are often considered the same group disease, based on the similar clinical behavior and possibly sharing the similar pathogenesis [12-18]. In 1996, McGuire and colleagues conducted a clinical trial to set up the standard therapy for patients with ETOC, including primary debulking surgery (PDS), also called primary cytoreductive surgery (PCS) plus adjuvant triweekly paclitaxel and cisplatin therapy [19]. The following studies have confirmed the superiority of survival benefits using paclitaxel into cisplatin-based regimens in place of cyclophosphamide for the treatment of patients with ETOC and PPSC [20,21]. Additionally, carboplatin in place of cisplatin in this combination has statistically significantly reduced the cisplatin-related potential toxicity of neural and renal systems and ameliorated the cisplatin-related high emetic effects [22-25].

  • Current Standard of Treatment

According to the National Comprehensive Cancer Network (NCCN) guideless version 1.2020 Ovarian Cancer/Fallopian Tube Cancer/Primary Perineal Cancer, principles of systemic therapy regimens for advanced-stage high-grade serous ETOC and PPSC included three parts [26]. Preferred regimens [26] include (1) the combination of intravenous (IV) paclitaxel 175 mg/m2 and IV carboplatin area under the curve ranging from 5 mg/mL per min (AUC 5) to 6 mg/mL per min (AUC 6) every three weeks (Q3W, also called triweekly) for 3-6 cycles [22-25]; (2) IV paclitaxel 175 mg/m2 day 1 followed by IV carboplatin AUC 6 Day 1 and bevacizumab 7.5 mg/kg Day 1 of cycle 1 Q3W for 5-6 cycles plus additional 12 cycles of bevacizumab Q3W for maintenance therapy; (3) IV paclitaxel 175 mg/m2 followed by IV carboplatin AUC 6 Day 1 Q3W for 6 cycles adding with bevacizumab 15 mg/kg which starts on Day 1 of cycle 2, Q3W for up to a total of 22 cycles [27-31].

Other recommended regimens [26] include the following four regimens, such as (1) IV paclitaxel 60 mg/m2followed by IV carboplatin AUC 2 every week (weekly) [32,33], (2) IV docetaxel 60-75 mg/m2 and IV carboplatin AUC 5-6 Q3W [34,35], (3) IV carboplatin AUC 5 and IV pegylated liposomal doxorubicin 30 mg/m2 every four weeks (Q4W) [36-38], and (4) IV paclitaxel 80 mg/m2 weekly and IV carboplatin AUC 5 to AUC 7 triweekly [39-42].

The final part, which is specially useful in certain circumstances, such as those patients having optimally PCS stage II-IV diseases [26], is the combination of IV paclitaxel 135 mg/m2 Day 1, intraperitoneal (IP) cisplatin 75-100 mg/m2 Day 2 and IV paclitaxel 60 mg/m2 Day 8 Q3W for 6 cycles [43,44]. Although there are so many recommendations available as shown above, now, the most frequently acceptable standard chemotherapy for ETOC and PPSC is still the use of combination of triweekly carboplatin and paclitaxel [45-51].

However, the prognosis of patients with ETOC and PPSC is still disappointing, because of vague or free symptoms of patients, often misdiagnosed as less deadly gastro-enteral tract problems, and absence of an effective screening programs, and in addition, heterogeneous in nature and different clinical development, accounting for late diagnosis in its advanced stages [52-55]. All result in a therapeutic challenge, contributing to low cure rate and high mortality rate, [1-5,19-25]. A median progression-free survival (PFS) is 16-21 months and an overall survival (OS) is 32-57 months [1-5,19-25]. Therefore, an improvement of therapeutic effect as well as a prolongation of PFS and/or OS is urgently needed. 

Based on relative worse prognosis of advanced-stage ETOC and PPSC, many new modalities and strategies have been developed recently [56-59]. Although some of them have been already recommended by updated NCCN guidelines for ETOC and PPSC treatment [4,26], they are not widely used in the routine clinical practice [60]. Among these, induction chemotherapy (neoadjuvant chemotherapy- NACT) using either standard form of paclitaxel and carboplatin or its modification form, including dose-dense or high-dose chemotherapy, and in additional adding bevacizumab, has become more and more popular, especially for those patients who are not candidates for immediate PCS [61-69].

In addition, the changed delivery route or warm-up of chemotherapy agents (IP administration or IP hyperthermia therapy) has been also accepted in the certain clinical situations [43,44,70-75]. Moreover, advancing drug development, including poly(adenosine diphosphate (ADP)-ribose) polymerase inhibitors (PARPi) as well as small molecules targeting various kinds of signaling pathway has shown the dramatic improvement in ETOC and PPSC treatment [76-87]. Finally, immune checkpoint inhibitors or immune system modulators has also provided a chance for ETOC and PPSC patients, although the therapeutic effect is debated [88,89]. However, there is no doubt that the high cost of the aforementioned new therapeutic approaches will limit the clinical use in routine [90], and in addition, the compliance may be compromised by the requirement of maintenance or continuous long-term therapy in some drugs.

  • Gap in Knowledge of Current Standard of Therapy

Based on the negative impact on the high cost of many targeted therapies and the poor compliance of long-term maintenance therapy in certain population, as well as no differences or improvement of clinical outcomes in additional drugs added to platinum/taxane combination or the use of different platinum doublets [91], dose-dense platinum/taxane might be a good alternative, because of the similar duration of completing therapy and acceptable cost expense. In addition, some evidence also favored this regimen as a feasible front-line chemotherapy for patients after PCS based on the possible PFS benefits and considerably robust cost-effectiveness [92,93].

In theory, the maximal tolerated doses of chemotherapy may yield the highest cytotoxicity to tumors, with subsequent higher cure rates [94]. However, such treatment may need a longer treatment-free period to wait normal host cells for recovery [94]. Among these, hematopoietic progenitors might be the best example [94]. Without this treatment-free period, some catastrophic and life-threatened conditions might occur. During the holiday of drug, cancer cells and cells in tumor microenvironment may also re-grow, possibly resultant development of aggressive behavior and chemo-resistance clone of tumor cells. All decided a rule of thumb to design most chemotherapy combinations to be repeated every three or four weeks [95]. The triweekly carboplatin plus paclitaxel regimen is just followed by the aforementioned rule, and this protocol is still one of the best-known and golden-standard regimens in the management of EOC patients after PCS.

However, this “golden-standard” therapy only provides the relatively short PFS and unsatisfactory OS in these advanced-stage EOC patients, making the urgent needs of application of new strategies to improve therapeutic outcome and prolong the patients’ life. Since the use of 3-drug combinations and maintenance chemotherapy after the front-line chemotherapy is not recommended because of no survival benefits but significantly increased adverse events (AEs) and possibly compromising quality of life [50,51,59], another strategy without adding any new agents or targeted therapy is pursued [45]. Therefore, if single dose of treatment is decreased in their dosage, the holiday of drugs can be shortened. Dose-dense weekly prescription of drugs may fulfill the aforementioned requirement.

  • Rationale of Dose-Dense Therapy

Dose-dense weekly chemotherapy is based on the concept to shorten timing of recycling by Goldie-Coldman’s hypothesis and Norton-Simon model [96-98], providing the rationale, such as a reducing interval between treatment, an increasing duration of chemotherapy exposure, and an increasing dose-intensity or dose-density of cytotoxic agents, and offering a chance that tumor regrowth could be decreased by this approach [99-102]. In addition, dose-dense chemotherapy can not only preserve the immune system but also promote the treatment-mediated tumor-specific immunity, especially the antitumor kill cluster of differentiation 8+ (CD8+) T-cell response [103]. Therefore, hematological markers have been evaluated to try to establish their predictive value [104-107]. Dose intensity strategies include increasing dose per cycle, decreasing cycle interval (dose densification) and decreasing interval plus increasing dose [108]. However, the latter two strategies involved the shortening the gap between chemotherapy which has been proven decisive in some neoplasms, such as breast cancer [99], and of most importance, this more frequent administration of cytotoxic agents within the short interval has also decreased chemotherapy-related death associated with the use of massive doses, such as infection or sepsis caused by severe neutropenia or spontaneous hemorrhage caused by severe thrombocytopenia [49].

Dr. Muggia proposed a very interesting hypothesis to show the potential benefits of dose-dense sequential treatment designs for first line, such as an initial dose-dense cisplatin and following dose-dense paclitaxel based on unpublished data from GOG-132 and ICON3 (Gynecologic Cancer InterGroup (GCIG) International Collaboration on Ovarian Neoplasms 3) in 2003 [109]. In the next year (2004), the same author found that increasing the dose of cisplatin above a certain threshold is not recommended in EOC patients because the greater toxicity with higher dose of cisplatin was found [110], although some studies did not support a significant increase toxicity after weekly dose-dense cisplatin treatment (median dose intensity with 32 or 45 mg/m2/week) [108]. However, the therapeutic effect of dose-dense platinum is still controversial. Some are in favor [111,112], but many are against the resultant benefits of dose-dense platinum [108,113-116]. Biweekly dose-dense carboplatin (AUC5) combined with paclitaxel (175 mg/m2) is not feasible based on dose limiting toxicity, even though GCSF was used for support [116]. Previous studies that the main determinant of dose-dense treatment response was not achieved by the relative dose intensity of platinum itself [108]. Indeed, one report showed that the administration of dose up to cisplatin of 25 mg/m2/week might reach the plateau of dose-response curve [113].

By contrast, the toxicity of dose-dense taxane seemed to be well tolerated because of absence of relevant toxicity [94]. A pharmacokinetic study revealed that weekly paclitaxel without interruption can be administered at doses of 110 mg/m2, while neutropenia precluded scheduled administration of dose ≥ 130 mg/m2 [94,117]. In theory, weekly dose-dense paclitaxel provided the better cytotoxicity to tumors, through more sustained exposure, limiting the emergence of tumors resistant to chemotherapy, enhancing the apoptotic and antiangiogenic effect, and improving therapeutic index [94,118-120], although single use of dose-dense paclitaxel for the first-line chemotherapy for the treatment of patients with ETOC seemed to be not approved by clinical trials [109,110].

Therefore, the application of combination of taxane and platinum in ETOC has been conducted. In the management of all chemosensitive epithelial cancers, combination chemotherapy treatment has provided significant survival benefits compared to single agent chemotherapy when applied as initial therapy, and there is no doubt that the treatment of ETOC is similar in this regard [121]. Studies showed that combination of taxane and platinum compound appears both to attenuate the toxicity of the platinum compound and to facilitate the delivery of full dose on schedule and this full dose on schedule showed the strong correlation with patient outcomes [110]. Since both platinum and paclitaxel are the essential cytotoxic agents in ETOC patients after PCS, either any one or both can be administered as dose-dense protocol. Furthermore, dose-dense can be classified according to the types, including semiweekly dose-dense, in which paclitaxel was given weekly and carboplatin was given triweekly, and weekly dose dense, in which both paclitaxel and carboplatin were given weekly [47]. An accumulation dose of paclitaxel is 240-270 mg/m2 with separating into 80-90 mg/m2 per week compared to a single use of 175-180 mg/m2 every three weeks. By contrast, the dosage of carboplatin seems to be consistent with accumulation dose as AUC 5-7, regardless of administration triweekly (a single use of AUC 5-7) or weekly (AUC 2).

  • Previous studies for Dose-Dense Therapy

Since the rational of dose-dense therapy supports the potential benefits for patients with advanced stage ETOC and PPSC, at least four large randomized clinical trials were conducted to test the hypothesis [32,33,39-41]. The Japanese Gynecologic Oncology Group (JGOG) 3016, named as JGOG 3016, provided a strong evidence to show the survival benefits in the combination regimen including dose-dense weekly paclitaxel (80 mg/m2) and triweekly carboplatin (AUC 6), which not only offered a longer PFS but also a significantly prolonged OS than those in standard triweekly combination of paclitaxel and carboplatin treatment [32,33].

By contrast, this promising result cannot be reproducible in other three trials conducted in Western countries. The results of an NRG Oncology (the National Cancer Institute Cooperative Group Program plus the Radiation Therapy Oncology plus Gynecology Oncology Group)/GOG Study showed the similar PFS and OS in dose-dense regimen with adding 15 mg/kg bevacizumab treatment compared to the standard triweekly combination therapy [72]. MITO-7 (Multicentre Italian Trials in Ovarian cancer 7) showed that there was no statistically significant difference of median PFS between dose-dense weekly combination and standard triweekly combination therapy [32]. GOG-0262 showed no statistical difference of the median PFS between dose-dense weekly paclitaxel and triweekly carboplatin with/without adding 15 mg/kg bevacizumab treatment, respectively, and standard triweekly combination therapy [75]. The results from an ICON8 showed that the restricted mean PFS was not statistically significant different among the three groups (dose-dense or standard groups) [33].

A recent meta-analysis conducted by Marchetti and colleagues further updated these four randomised controlled trials containing 3698 patients, and found that dose-dense chemotherapy did not have a statistically significant benefit of PFS (HR 0.92, 95% CI 0.81-1.04) [47]. Additionally, the results were also no change with HR 1.01 (95% CI 0.93-1.10) and 0.82 (95% CI 0.63-1.08), respectively, even though the analysis was limited to both weekly and semi-weekly dose-dense data [47]. Therefore, the authors believed that conventional triweekly combination chemotherapy is still one of the golden-standard treatments of patients with advanced stage ETOC [47].

Why the survival benefits are only apparent in the certain population, such as Japanese or others? Although the real reasons associated with conflicted results are uncertain, genomic background of race, tumor heterogeneity, administration schedule, cisplatin or carboplatin, duration to complete postoperative adjuvant chemotherapy may be attributable. Among these, we believe that platinum compound may play a certain role in this discrepancy.

In order to evaluate the effects and safety of dose-dense of paclitaxel combining with either cisplatin or carboplatin, the following retrospective study was conducted. Patients with FIGO IIIC serous-type ETOC and PPSC treated either with triweekly cisplatin (20 mg/m2) plus dose-dense weekly paclitaxel (80 mg/m2) or triweekly carboplatin (AUC 5) plus dose-dense weekly paclitaxel (80 mg/m2) as postoperative adjuvant chemotherapy after PCS were retrospectively reviewed. However, nearly all dose-dense paclitaxel-platinum combination chemotherapy regimens used as the front-line postoperative adjuvant therapy are focused on the dose-dense paclitaxel, without consideration of platinum compounds or dose-dense both agents.

Discussion

In my opinion this section seems to be too speculative and repetitive in some aspects. You could summarize some concepts.

Response: Thank you very much and we have significantly shortened this part and we present it in brief and precise. Please read: 

4. Discussion

4.1. Main findings

The main findings of the current study showed that both regimens could be successfully used for the treatment of patients with FIGO IIIC ETOC or PPSC, offering 30 and 25 months of median PFS in paclitaxel-cisplatin and paclitaxel-carboplatin groups, respectively. Median OS was 58.5 months in paclitaxel-cisplatin group and 55.0 months in paclitaxel-carboplatin group. Although either PFS or OS did not reach the statistically significant difference, it seemed to be little better in patients treated with paclitaxel-cisplatin regimen compared to those treated with paclitaxel-carboplatin regimen. However, to compare the frequency and severity of AEs, we found that patients in paclitaxel-carboplatin group had more AEs than those in paclitaxel-cisplatin group did, including a higher risk of neutropenia and grade 3/4 neutropenia, and the need of the longer period to complete the front-line chemotherapy. The need of the longer period to finish the dose-dense therapy seemed to be associated with worse outcome of patients. Further analysis showed that the period between the first-time chemotherapy to the last dose (6 cycles) of chemotherapy > 21 weeks was associated with worse prognosis in patients compared to that ≤21 weeks, with hazard ratio (HR) of 81.24 for PFS and 9.57 for OS. As predicted, suboptimal debulking surgery (>1 cm) also contributed to the worse outcome than optimal debulking surgery (≤1 cm) with HR of 14.38 for PFS and 11.83 for OS.

4.2. Summary of Studies Addressing Dose-Dense therapy: Survival Outcome (Table 5)

Some early phase studies have also been conducted to test the tolerability and efficacy of dose-dense weekly paclitaxel (80 mg/m2) and triweekly carboplatin (AUC 5) with other agents [132,133]. For example, one phase II study modified a dose-dense paclitaxel (80 mg/m2) and every 4-week carboplatin (AUC 5) for the treatment of advanced-stage ETOC patients and found that median PFS and OS were 22.5 and 31.5 months, respectively [132]. Another phase II study with adding bevacizumab in dose-dense chemotherapy (weekly paclitaxel 80 mg/m2 and triweekly carboplatin AUC 5) showed the median PFS was 16.9-22.4 months in advanced stage ETOC patients [133].

Based on the impressive survival benefits of the earlier reports, the prospective, randomised trials may be the best method to test the efficacy and safety of dose-dense chemotherapy in the management of ETOC patients. The first large study, named as the JGOG 3016 has shown the survival benefits in dose-dense regimen containing weekly paclitaxel (80 mg/m2) and triweekly carboplatin (AUC 6), which not only offered a longer PFS (median PFS of 28.2 months, 95% CI 22.3-33.8 months) but also a significantly prolonged OS (median OS of 100.5 months, 95% CI 65.2-∞ months) than those in standard triweekly paclitaxel-carboplatin regimen [32,33].

The results of an NRG Oncology/GOG Study showed the similar PFS (the median PFS of 24.9 months) and OS (the median OS of 75.5 months) in dose-dense weekly paclitaxel (80 mg/m2) and triweekly carboplatin (AUC 6) with adding 15 mg/kg bevacizumab treatment compared to the standard triweekly combination therapy [72].

MITO-7 has shown that there was no statistically significant difference of median PFS between dose-dense weekly combination (paclitaxel 60 mg/m2 plus carboplatin AUC 2) and standard triweekly combination therapy (18.3 months [95% CI 16.8-20.9 months] versus 17.3 months [95% CI 15.2-20.2 months]) with HR of 0.96, 95% CI 0.80-1.16 [32].

GOG-0262 showed the median PFS of 14.9 and 14.2 months in dose-dense weekly paclitaxel (80 mg/m2) and triweekly carboplatin (AUC 6) with/without adding 15 mg/kg bevacizumab treatment, respectively, without statistically significant difference compared to the standard triweekly combination therapy [41].

The results from an ICON8 have shown that the median PFS was 20.8 months (95% CI 11.9-59.0 months) and 21.0 months (95% CI 12.0-54.0 months) in dose-dense weekly paclitaxel (80 mg/m2) and triweekly carboplatin (AUC 5 or 6) and in dose-dense weekly paclitaxel (80 mg/m2) and weekly carboplatin (AUC 2) regimen, respectively, and none of both was statistically significant different from the 17.7 months of the median PFS in standard-dose triweekly paclitaxel and triweekly carboplatin (AUC) treatment [33].

Although evidence might be much more powerful when the results were obtained from the prospective, randomised trials, the management of the individual patient who does not fulfill the criteria of a clinical trial or is enrolled into the clinical trial will be debated continuously. In addition, the strict criteria of a clinical trial, such as inclusive and exclusive criteria may not be reflective in real situations of patients. Therefore, clinical practices are not uniform, retrospective evaluations may partly reveal a real-world clinical practice. For example, elder population is often excluded in the clinical trials. Dr. Bun and colleagues found dose-dense weekly paclitaxel plus triweekly carboplatin was feasible for elderly patients although severe neuropathy might occur frequently compared with that in younger group [134]. In fact, elder population might have a higher risk to receive nonstandard chemotherapy [135].

Table 5. Summary of treatment efficacy and safety of dose-dense weekly paclitaxel plus platinum compounds in the management of patients with epithelial tubo-ovarian cancer and primary peritoneal serous carcinoma.

Authors

Stage

n

Regimen

PFS

OS

Prospective randomized trials

Katsumata [40]

II–IV

312

P 80 mg/m2 (D1,8,15),

C 6 (D1)

28.2 (M)

100.5 (M)

Pignata [32]

  IC-IV

406

P 60 mg/m2 (D1,8,15),

C 2 (D1,8,15)

18.3 (M)

Chan [41]

II–IV

340

P 80 mg/m2 (D1,8,15),

C 6 ±BEV (D1)

14.7 (M)

-

55

P 80mg/m2 (D1,8,15),

C 6 (D1)

14.2 (M)

-

Clamp [33]

I-IV

P 80 mg/m2 (D1,8,15),

C 5,6 ± BEV (D1)

20.8 (M)

P 80 mg/m2 (D1,8,15),

C 2 (D1,8,15) ± BEV (D1)

21.0 (M)

Walker [72]

II–IV

521

P 80 mg/m2 (D1,8,15),

C 6 +BEV (D1)

24.9 (M)

75.5 (M)

Retrospective study, including phase II study

Abaid [132]

III-IV

88

P 80 mg/m2 (D1,8,15),

C 5 (D1), stop one week

22.5 (M)

31.5 (M)

Fleming [133]

III-IV

33

P 80 mg/m2 (D1,8,15),

C 5 + BEV (D1)

16.9-22.4 (M)

Murphy [102]

III

38

P 80 mg/m2 (D1,8,15),

C 5 (D1)

31.3 (m)

54.5 (m)

Boraska Jelavić [105]

I-IV

43

P 80 mg/m2 (D1,8,15),

C 5 (D1)

20-24 (M)

Rettenmaier [78]

I-IV

100

P 80 mg/m2 (D1,8,15),

C 5 (D1)

27.6 (M)

Cheng [101]

IIIC-IV

32

P 80 mg/m2 (D1,8,15),

Cisplatin 20 mg/m2 (D1)

27.0 (M)

56 (m)

Current study

IIIC

18

P 80 mg/m2 (D1,8,15),

Cisplatin 20 mg/m2 (D1)

30.0 (M)

58.5 (M)

22

P 80 mg/m2 (D1,8,15),

C 5 (D1)

25.0 (M)

55.0 (M)

Dose-dense chemotherapy is limited on the intravenous route. RCT: randomized control trial, containing co-administration of bevacizumab (BEV) 15 mg/kilograms at day 1 triweekly and patients treated with neoadjuvant chemotherapy; n: number of patients; PFS: median (M) or mean (m) progression-free survival calculated by months; OS: median or mean overall survival calculated by months; Stage: FIGO stage (International Federation of Gynecology and Obstetrics stage); P: paclitaxel; D: day; C 2, C 5 or C 6: carboplatin (AUC 2, AUC 5 or AUC 6: area under the curve 2, 5, or 6 mg/mL per minute).

4.3. Dose-Dense Therapy-Related Adverse Events: Prefer the Use of Low-Dose Cisplatin in Place of Carboplatin in Platinum-based Therapy

It is believed in long-term that the use of carboplatin in place of original cisplatin in the platinum compound-based paclitaxel doublet might provide a better quality of life (QOL), since this regimen, with carboplatin dosed using the Calvert formula, yielded convincing non inferior outcomes when compared with the prior, more toxic, cisplatin-paclitaxel regimen [136]. By contrast, bone marrow suppression, including toxicity of hematological system might be more apparent in carboplatin administration, although Boyd and Muggia proposed that carboplatin-paclitaxel therapy is generally safe, when this drug is properly dosed [136]. Occurrence of neutropenia is one of the most causes to postpone the therapy or modify (decrease) the dosage of therapeutic agent. One study from Japan in 2006 tried to determine the feasibility of docetaxel-cisplatin therapy Q3W compared with docetaxel-carboplatin therapy Q3W as front-line for ETOC patients [136]. The results, as predicted, showed that incidence of grade 4 neutropenia was much more higher in the docetaxel-carboplatin group than that in the docetaxel-cisplatin group (74% versus 39%), suggesting the feasibility of docetaxel-cisplatin combination therapy as front-line therapy for ETOC patients [136]. In addition, rationale of the use “dose-dense” in place of “standard-dose” attempts to decrease the incidence of chemotherapy-related severe hematotoxicity, such as infection or sepsis caused by severe neutropenia or spontaneous hemorrhage caused by severe thrombocytopenia [49]. In Taiwan, GCSF is not prescribed in routine and insurance does not cover the cost if GCSF is used as prophylactic role. Furthermore, the obstacle is still present in facing patients with chemotherapy-related thrombocytopenia [137-139].

Thrombocytopenia is principal consideration and the dose-limiting toxicity of carboplatin [140,141]. Although thrombocytopenia unlikely occurs in a chemotherapy-naïve patients, but this toxicity usually begins to appear after day 14 and is predictably cumulative, contributing to the consideration of a carboplatin dose reduction [140]. Due to aforementioned limitation, some gynecological oncologists favored the use of cisplatin as the drug in the platinum-based combination therapy, except for those patients with identified impaired renal function (eGFR < 60 ml/min). The current low-dose cisplatin plus dose-dense paclitaxel regimen was associated with lower rates of hematologic toxicities compared to the conventional dose-dense paclitaxel plus carboplatin regimen. The results from an ICON8 have also shown that patients treated with weekly regimen had increased grade 3 or 4 toxic effects, although these high-grade toxicities were predominantly uncomplicated [33] However, both regimens are carboplatin-based paclitaxel combination.

Consistent with more toxicity of the use of carboplatin in previous study [136], the incidence of grade 3/4 neutropenia was significantly lower in the cisplatin-paclitaxel arm than that in the carboplatin-paclitaxel arm (27.8% versus 77.3%, p = 0.002) in our study. In addition, other parameters of hematological examination, including grade 3/4 anemia and grade 3/4 thrombocytopenia also favored the cisplatin-paclitaxel regimen with incidence of 5.6% and none, respectively compared to 22.7% and 13.6% in the carboplatin-paclitaxel regimen in the current study. We believed that this carboplatin-paclitaxel combination therapy-related hematotoxicity might explain more patients in the carboplatin-paclitaxel arm had a prolongation of the period between the initial first time of chemotherapy and the end of final chemotherapy (six cycles of chemotherapy) than those in the cisplatin-paclitaxel arm did (45% versus 11.1%, p = 0.018). Further survival analysis showed this prolongation of therapeutic period might be associated with bad outcome of FIGO IIIC ETOC and PPSC patients.

By contrast, the cisplatin-related renal, neural, and gastrointestinal system toxicity was lower in the current study. The main reason is the use of low-dose cisplatin (20 mg/m2) as a platinum compound in the platinum-paclitaxel combination therapy.        

4.4. The Benefits of Maximal Cytoreductive Surgery and The Consideration of the Location of Residual Tumors

In agreement with well-known concept [142-146], a rapid reduction of the tumor burden is the guarantee of the successful treatment in cancer patients. Nearly all studied have confirmed that maximal cytoreduction surgery is the most critical step in the management of patients with advanced-stage ETOC or PPSC [142-146]. According to the definition of the Gynecologic Oncology group (GOG) for 'optimal' as having residual tumor nodules each, which should be ≤ 1 cm in size, Cochrane review revealed that outcome of ETOC patients with residual disease ≤ 1 cm is better than that with residual disease > 1 cm [143], although it is well known that PCS to no gross residual tumor might be associated with the longest PFS and OS [144,145]. One study showed an interesting finding that patients still had diverse outcomes despite undergoing optimal PCS (≤ 1 cm residual disease), in which it demonstrated that patients with tumor limited in a single anatomic location had better median PFS and OS than patients with tumors widely distributed in multiple anatomic locations [146]. In our current study, we also tested the aforementioned findings. We found that patients with suboptimal PCS with a residual tumor > 1 cm indeed had a worse prognosis with a hazard ratio (HR) of 14.38 (95% confidence interval [CI] 4.18-49.46) in PFS and a HR of 11.83 (95% CI 1.48-94.72) in OS, comparing with those patients who could achieve optimal PCS with a residual tumor in size less than 1cm, supporting the rationale that small-volume tumors are more prone to clearance [61]. However, the widespread of tumor seemed to be not an independent risk factor for worse prognosis. We found that there was no statistically significant difference of outcome in patients whose residual tumors in localized anatomical site or in multiple anatomical sites.

4.5. The Strength and The Limitation

The current study has the following strength. All patients were serous type ETOC and PPSC with FIGO stage IIIC. The follow-up period was long enough with nearly 5 years (a mean follow-up period of 55 months). All patients were treated with by one senior doctor (P.-H.W). All suggested study population was homogeneous.

There are some limitations in the current study. Retrospective data collection in nature may not be strong to show the evidence. However, as shown above, it is a real-world clinical practice. In addition, only patients who underwent PCS with surgical confirmation of FIGO IIIC serous-type ETOC and PPSC and complete 6 cycles of dose-dense paclitaxel-platinum compounds were included, contributing to the risk of selection bias. Furthermore, number in the current study was small in each arm, contributing to wide confidence intervals. It demonstrated the higher risk of low precision of the estimates. Therefore, the findings of the current study should be used in caution. Finally, new therapeutic strategies, such as maintenance therapy or NACT or intraperitoneal therapy were not evaluated in the current report. However, we believe none of them will impede the value of current study.

5. Conclusions

Our study found that paclitaxel might be one of the most critical components for the successful administration of dose-dense adjuvant chemotherapy. In addition, on-schedule and shortening of the entire therapeutic period are a critical step for the survival benefits of patients who underwent a postoperative adjuvant dose-dense paclitaxel plus platinum compounds regimen. Of course, maximal efforts should be done to eradicate the tumor as much as possible. Although recent evidences highlighted the value of precise medicine, personalized and individual therapy along with targeted therapy for cancer patients [147-151], the net health benefits (NHBs), cost-effectiveness or quality of life may reflect the relative value of treatment options in EOC [100, 152-157]. With low risk of AEs, and shortening therapeutic period, and of most importance, without compromising therapeutic effects, low-dose triweekly cisplatin plus dose-dense weekly paclitaxel might be associated with the better NHBs, more cost-effectiveness and a better QOL compared to carboplatin-based dose-dense paclitaxel regimen. More studies are welcome to test our findings.

Reviewer 2 Report

This is a small size clinical trial. The introduction and discussion although present important material - are very long. The reader will appreciate if the introduction and discussion are organized as following:

Introduction subtitles:

  1. Current standard of treatment
  2. Gap in knowledge
  3. Specific study features that address the gap in current knowledge

Study design:

It is not clear whether patients were randomized to each treatment and whether informed consent was obtained. The criteria on how each patient was assigned to a specific treatment is absent.

Discussion subtitles:

  1. Main findings
  2. Comparison and discussion of finding A against the results of other studies and theoretical background
  3. The same for finding B, etc.

One of the limitations of this study is small sample size, which resulted in wide confidence intervals demonstrating low precision of the estimates. This must be stated and the results should be qualified as indicative or suggestive.

Overall, this manuscript will benefit from better more clear organization of the material.  

Author Response

Manuscript ID IJERPH-731096R1 and title: Compare Dose-Dense Paclitaxel Plus either Standard-Dose Carboplatin or Low-Dose Cisplatin as the Front-Line Regimen in the Management of FIGO IIIC Serous-Type Ovarian, Fallopian Tube, and Primary Peritoneal Cancer

Peng-Hui Peter Wang, M.D., Ph.D.,

Professor and Director

Department of Obstetrics and Gynecology,

Taipei Veterans General Hospital, and National Yang-Ming University School of Medicine

201, Shih-Pai Road, Section 2, Taipei 112, TAIWAN

Telephone: 886-2-2875-7566; Fax: 886-2-5570 2788

E-mail: phwang@vghtpe.gov.tw; pongpongwang@gmail.com

Special Issue: Gynecological Cancer

Date: 20 March 2020

Dear Editors and Reviewer II

Thank you very much for your information by email on date of 18 March 2020. We are very happy to have a chance to revise our article with manuscript number of IJERPH-731096R1 and a title as “Compare Dose-Dense Paclitaxel Plus either Standard-Dose Carboplatin or Low-Dose Cisplatin as the Front-Line Regimen in the Management of FIGO IIIC Serous-Type Ovarian, Fallopian Tube, and Primary Peritoneal Cancer, which is submitted to the Special Issues “Gynecological Cancer” of the International Journal of Environmental Research and Public Health (ISSN 1660-4601) for your consideration.

We followed the suggestions made by experts (reviewers and academic editor) International Journal of Environmental Research and Public Health Editorial Office to revise this manuscript, which includes an itemized, point-by-point response to each reviewer, including the reviewer's original comment(s). We specify the changes made to address each of their concerns and we also include the changes made in the response and indicate the locations in the manuscript. We address each reviewer’s comment using the reviewer's number. Please see the following pages at the end of this letter.

We also agree to pay to have our article published under the Open Access option if the article fulfills the criteria for publication.

All co-authors have approved this manuscript and agreed to submit this original article to the International Journal of Environmental Research and Public Health. We also agree to pay to have our article published under the Open Access option if the article fulfills the criteria for publication. That is to say that we agree that the article processing charges (APC) of Swiss Francs apply to accepted papers. We totally agree the following: this is an open access article distributed under the Creative Commons Attribution License 4.0, which permits unrestricted use, distribution, and reproduction in any medium, provided the original work is properly cited.

All authors certify that this manuscript is a unique submission and is not being considered for publication by any other source in any medium. Further, the manuscript has not been published, in part or in full, in any form.

We appreciate your kindness for further considering it for publication. We are looking forward to hearing from you soon.

Very sincerely yours,

Peng-Hui Wang, MD, PhD

Chen-Yu Huang, (eu.huang501@gmail.com); Min Cheng, (alchemist791025@gmail.com); Na-Rong Lee, (nllee@vghtpe.gov.tw); Hsin-Yi Huang (sweethsin509@gmail.com); Wen-Ling Lee, (johnweiwang@gmail.com); Wen-Hsun Chang (whchang818@gmail.com)

Response to reviewer II’s Comments

Open Review

English language and style

( ) Extensive editing of English language and style required  
( ) Moderate English changes required  
(x) English language and style are fine/minor spell check required  
( ) I don't feel qualified to judge about the English language and style 

Response: Thank you very much for your favorable consideration, and text has been already carefully edited.

Yes

Can be improved

Must be improved

Not applicable

Does the introduction provide sufficient background and include all relevant references?

X

Is the research design appropriate?

X

Are the methods adequately described?

X

Are the results clearly presented?

X

Are the conclusions supported by the results?

X

Response: Thank you very much for your favorable consideration. We have followed your valuable comments to re-organize the current study and the response to every comment is shown below point-by-point. Thank you again. Please read:

Comments and Suggestions for Authors

This is a small size clinical trial. The introduction and discussion although present important material - are very long. The reader will appreciate if the introduction and discussion are organized as following:

Introduction subtitles:

  1. Current standard of treatment
  2. Gap in knowledge
  3. Specific study features that address the gap in current knowledge

Response: Thank you very much for your valuable recommendation. We have followed your comments to re-organize the current study and the response to every comment is shown below point-by-point. Thank you again. Please read:

1. Introduction

Epithelial tubo-ovarian cancer (ETOC) is the deadliest cancer among women placing with 4th place of all the fetal diseases among women and ranking 7th of most common cancer in women cancer and women’s cancer-related death in Taiwan and globally in 2018 [1-6]. Serous-type is the most common subtype among EOCs in general; however, endometriosis-associated EOCs, such as clear cell type or endometrioid type are relatively common in certain populations, including Taiwan and Japan [7-11]. Clinically, the serous-type EOC and primary peritoneal serous carcinoma (PPSC) and primary Fallopian tube cancer (PFTC) are often considered the same group disease, based on the similar clinical behavior and possibly sharing the similar pathogenesis [12-18]. In 1996, McGuire and colleagues conducted a clinical trial to set up the standard therapy for patients with ETOC, including primary debulking surgery (PDS), also called primary cytoreductive surgery (PCS) plus adjuvant triweekly paclitaxel and cisplatin therapy [19]. The following studies have confirmed the superiority of survival benefits using paclitaxel into cisplatin-based regimens in place of cyclophosphamide for the treatment of patients with ETOC and PPSC [20,21]. Additionally, carboplatin in place of cisplatin in this combination has statistically significantly reduced the cisplatin-related potential toxicity of neural and renal systems and ameliorated the cisplatin-related high emetic effects [22-25].

  • Current Standard of Treatment

According to the National Comprehensive Cancer Network (NCCN) guideless version 1.2020 Ovarian Cancer/Fallopian Tube Cancer/Primary Perineal Cancer, principles of systemic therapy regimens for advanced-stage high-grade serous ETOC and PPSC included three parts [26]. Preferred regimens [26] include (1) the combination of intravenous (IV) paclitaxel 175 mg/m2 and IV carboplatin area under the curve ranging from 5 mg/mL per min (AUC 5) to 6 mg/mL per min (AUC 6) every three weeks (Q3W, also called triweekly) for 3-6 cycles [22-25]; (2) IV paclitaxel 175 mg/m2 day 1 followed by IV carboplatin AUC 6 Day 1 and bevacizumab 7.5 mg/kg Day 1 of cycle 1 Q3W for 5-6 cycles plus additional 12 cycles of bevacizumab Q3W for maintenance therapy; (3) IV paclitaxel 175 mg/m2 followed by IV carboplatin AUC 6 Day 1 Q3W for 6 cycles adding with bevacizumab 15 mg/kg which starts on Day 1 of cycle 2, Q3W for up to a total of 22 cycles [27-31].

Other recommended regimens [26] include the following four regimens, such as (1) IV paclitaxel 60 mg/m2followed by IV carboplatin AUC 2 every week (weekly) [32,33], (2) IV docetaxel 60-75 mg/m2 and IV carboplatin AUC 5-6 Q3W [34,35], (3) IV carboplatin AUC 5 and IV pegylated liposomal doxorubicin 30 mg/m2 every four weeks (Q4W) [36-38], and (4) IV paclitaxel 80 mg/m2 weekly and IV carboplatin AUC 5 to AUC 7 triweekly [39-42].

The final part, which is specially useful in certain circumstances, such as those patients having optimally PCS stage II-IV diseases [26], is the combination of IV paclitaxel 135 mg/m2 Day 1, intraperitoneal (IP) cisplatin 75-100 mg/m2 Day 2 and IV paclitaxel 60 mg/m2 Day 8 Q3W for 6 cycles [43,44]. Although there are so many recommendations available as shown above, now, the most frequently acceptable standard chemotherapy for ETOC and PPSC is still the use of combination of triweekly carboplatin and paclitaxel [45-51].

However, the prognosis of patients with ETOC and PPSC is still disappointing, because of vague or free symptoms of patients, often misdiagnosed as less deadly gastro-enteral tract problems, and absence of an effective screening programs, and in addition, heterogeneous in nature and different clinical development, accounting for late diagnosis in its advanced stages [52-55]. All result in a therapeutic challenge, contributing to low cure rate and high mortality rate, [1-5,19-25]. A median progression-free survival (PFS) is 16-21 months and an overall survival (OS) is 32-57 months [1-5,19-25]. Therefore, an improvement of therapeutic effect as well as a prolongation of PFS and/or OS is urgently needed. 

Based on relative worse prognosis of advanced-stage ETOC and PPSC, many new modalities and strategies have been developed recently [56-59]. Although some of them have been already recommended by updated NCCN guidelines for ETOC and PPSC treatment [4,26], they are not widely used in the routine clinical practice [60]. Among these, induction chemotherapy (neoadjuvant chemotherapy- NACT) using either standard form of paclitaxel and carboplatin or its modification form, including dose-dense or high-dose chemotherapy, and in additional adding bevacizumab, has become more and more popular, especially for those patients who are not candidates for immediate PCS [61-69].

In addition, the changed delivery route or warm-up of chemotherapy agents (IP administration or IP hyperthermia therapy) has been also accepted in the certain clinical situations [43,44,70-75]. Moreover, advancing drug development, including poly(adenosine diphosphate (ADP)-ribose) polymerase inhibitors (PARPi) as well as small molecules targeting various kinds of signaling pathway has shown the dramatic improvement in ETOC and PPSC treatment [76-87]. Finally, immune checkpoint inhibitors or immune system modulators has also provided a chance for ETOC and PPSC patients, although the therapeutic effect is debated [88,89]. However, there is no doubt that the high cost of the aforementioned new therapeutic approaches will limit the clinical use in routine [90], and in addition, the compliance may be compromised by the requirement of maintenance or continuous long-term therapy in some drugs.

  • Gap in Knowledge of Current Standard of Therapy

Based on the negative impact on the high cost of many targeted therapies and the poor compliance of long-term maintenance therapy in certain population, as well as no differences or improvement of clinical outcomes in additional drugs added to platinum/taxane combination or the use of different platinum doublets [91], dose-dense platinum/taxane might be a good alternative, because of the similar duration of completing therapy and acceptable cost expense. In addition, some evidence also favored this regimen as a feasible front-line chemotherapy for patients after PCS based on the possible PFS benefits and considerably robust cost-effectiveness [92,93].

In theory, the maximal tolerated doses of chemotherapy may yield the highest cytotoxicity to tumors, with subsequent higher cure rates [94]. However, such treatment may need a longer treatment-free period to wait normal host cells for recovery [94]. Among these, hematopoietic progenitors might be the best example [94]. Without this treatment-free period, some catastrophic and life-threatened conditions might occur. During the holiday of drug, cancer cells and cells in tumor microenvironment may also re-grow, possibly resultant development of aggressive behavior and chemo-resistance clone of tumor cells. All decided a rule of thumb to design most chemotherapy combinations to be repeated every three or four weeks [95]. The triweekly carboplatin plus paclitaxel regimen is just followed by the aforementioned rule, and this protocol is still one of the best-known and golden-standard regimens in the management of EOC patients after PCS.

However, this “golden-standard” therapy only provides the relatively short PFS and unsatisfactory OS in these advanced-stage EOC patients, making the urgent needs of application of new strategies to improve therapeutic outcome and prolong the patients’ life. Since the use of 3-drug combinations and maintenance chemotherapy after the front-line chemotherapy is not recommended because of no survival benefits but significantly increased adverse events (AEs) and possibly compromising quality of life [50,51,59], another strategy without adding any new agents or targeted therapy is pursued [45]. Therefore, if single dose of treatment is decreased in their dosage, the holiday of drugs can be shortened. Dose-dense weekly prescription of drugs may fulfill the aforementioned requirement.

  • Rationale of Dose-Dense Therapy

Dose-dense weekly chemotherapy is based on the concept to shorten timing of recycling by Goldie-Coldman’s hypothesis and Norton-Simon model [96-98], providing the rationale, such as a reducing interval between treatment, an increasing duration of chemotherapy exposure, and an increasing dose-intensity or dose-density of cytotoxic agents, and offering a chance that tumor regrowth could be decreased by this approach [99-102]. In addition, dose-dense chemotherapy can not only preserve the immune system but also promote the treatment-mediated tumor-specific immunity, especially the antitumor kill cluster of differentiation 8+ (CD8+) T-cell response [103]. Therefore, hematological markers have been evaluated to try to establish their predictive value [104-107]. Dose intensity strategies include increasing dose per cycle, decreasing cycle interval (dose densification) and decreasing interval plus increasing dose [108]. However, the latter two strategies involved the shortening the gap between chemotherapy which has been proven decisive in some neoplasms, such as breast cancer [99], and of most importance, this more frequent administration of cytotoxic agents within the short interval has also decreased chemotherapy-related death associated with the use of massive doses, such as infection or sepsis caused by severe neutropenia or spontaneous hemorrhage caused by severe thrombocytopenia [49].

Dr. Muggia proposed a very interesting hypothesis to show the potential benefits of dose-dense sequential treatment designs for first line, such as an initial dose-dense cisplatin and following dose-dense paclitaxel based on unpublished data from GOG-132 and ICON3 (Gynecologic Cancer InterGroup (GCIG) International Collaboration on Ovarian Neoplasms 3) in 2003 [109]. In the next year (2004), the same author found that increasing the dose of cisplatin above a certain threshold is not recommended in EOC patients because the greater toxicity with higher dose of cisplatin was found [110], although some studies did not support a significant increase toxicity after weekly dose-dense cisplatin treatment (median dose intensity with 32 or 45 mg/m2/week) [108]. However, the therapeutic effect of dose-dense platinum is still controversial. Some are in favor [111,112], but many are against the resultant benefits of dose-dense platinum [108,113-116]. Biweekly dose-dense carboplatin (AUC5) combined with paclitaxel (175 mg/m2) is not feasible based on dose limiting toxicity, even though GCSF was used for support [116]. Previous studies that the main determinant of dose-dense treatment response was not achieved by the relative dose intensity of platinum itself [108]. Indeed, one report showed that the administration of dose up to cisplatin of 25 mg/m2/week might reach the plateau of dose-response curve [113].

By contrast, the toxicity of dose-dense taxane seemed to be well tolerated because of absence of relevant toxicity [94]. A pharmacokinetic study revealed that weekly paclitaxel without interruption can be administered at doses of 110 mg/m2, while neutropenia precluded scheduled administration of dose ≥ 130 mg/m2 [94,117]. In theory, weekly dose-dense paclitaxel provided the better cytotoxicity to tumors, through more sustained exposure, limiting the emergence of tumors resistant to chemotherapy, enhancing the apoptotic and antiangiogenic effect, and improving therapeutic index [94,118-120], although single use of dose-dense paclitaxel for the first-line chemotherapy for the treatment of patients with ETOC seemed to be not approved by clinical trials [109,110].

Therefore, the application of combination of taxane and platinum in ETOC has been conducted. In the management of all chemosensitive epithelial cancers, combination chemotherapy treatment has provided significant survival benefits compared to single agent chemotherapy when applied as initial therapy, and there is no doubt that the treatment of ETOC is similar in this regard [121]. Studies showed that combination of taxane and platinum compound appears both to attenuate the toxicity of the platinum compound and to facilitate the delivery of full dose on schedule and this full dose on schedule showed the strong correlation with patient outcomes [110]. Since both platinum and paclitaxel are the essential cytotoxic agents in ETOC patients after PCS, either any one or both can be administered as dose-dense protocol. Furthermore, dose-dense can be classified according to the types, including semiweekly dose-dense, in which paclitaxel was given weekly and carboplatin was given triweekly, and weekly dose dense, in which both paclitaxel and carboplatin were given weekly [47]. An accumulation dose of paclitaxel is 240-270 mg/m2 with separating into 80-90 mg/m2 per week compared to a single use of 175-180 mg/m2 every three weeks. By contrast, the dosage of carboplatin seems to be consistent with accumulation dose as AUC 5-7, regardless of administration triweekly (a single use of AUC 5-7) or weekly (AUC 2).

  • Previous studies for Dose-Dense Therapy

Since the rational of dose-dense therapy supports the potential benefits for patients with advanced stage ETOC and PPSC, at least four large randomized clinical trials were conducted to test the hypothesis [32,33,39-41]. The Japanese Gynecologic Oncology Group (JGOG) 3016, named as JGOG 3016, provided a strong evidence to show the survival benefits in the combination regimen including dose-dense weekly paclitaxel (80 mg/m2) and triweekly carboplatin (AUC 6), which not only offered a longer PFS but also a significantly prolonged OS than those in standard triweekly combination of paclitaxel and carboplatin treatment [32,33].

By contrast, this promising result cannot be reproducible in other three trials conducted in Western countries. The results of an NRG Oncology (the National Cancer Institute Cooperative Group Program plus the Radiation Therapy Oncology plus Gynecology Oncology Group)/GOG Study showed the similar PFS and OS in dose-dense regimen with adding 15 mg/kg bevacizumab treatment compared to the standard triweekly combination therapy [72]. MITO-7 (Multicentre Italian Trials in Ovarian cancer 7) showed that there was no statistically significant difference of median PFS between dose-dense weekly combination and standard triweekly combination therapy [32]. GOG-0262 showed no statistical difference of the median PFS between dose-dense weekly paclitaxel and triweekly carboplatin with/without adding 15 mg/kg bevacizumab treatment, respectively, and standard triweekly combination therapy [75]. The results from an ICON8 showed that the restricted mean PFS was not statistically significant different among the three groups (dose-dense or standard groups) [33].

A recent meta-analysis conducted by Marchetti and colleagues further updated these four randomised controlled trials containing 3698 patients, and found that dose-dense chemotherapy did not have a statistically significant benefit of PFS (HR 0.92, 95% CI 0.81-1.04) [47]. Additionally, the results were also no change with HR 1.01 (95% CI 0.93-1.10) and 0.82 (95% CI 0.63-1.08), respectively, even though the analysis was limited to both weekly and semi-weekly dose-dense data [47]. Therefore, the authors believed that conventional triweekly combination chemotherapy is still one of the golden-standard treatments of patients with advanced stage ETOC [47].

Why the survival benefits are only apparent in the certain population, such as Japanese or others? Although the real reasons associated with conflicted results are uncertain, genomic background of race, tumor heterogeneity, administration schedule, cisplatin or carboplatin, duration to complete postoperative adjuvant chemotherapy may be attributable. Among these, we believe that platinum compound may play a certain role in this discrepancy.

In order to evaluate the effects and safety of dose-dense of paclitaxel combining with either cisplatin or carboplatin, the following retrospective study was conducted. Patients with FIGO IIIC serous-type ETOC and PPSC treated either with triweekly cisplatin (20 mg/m2) plus dose-dense weekly paclitaxel (80 mg/m2) or triweekly carboplatin (AUC 5) plus dose-dense weekly paclitaxel (80 mg/m2) as postoperative adjuvant chemotherapy after PCS were retrospectively reviewed. However, nearly all dose-dense paclitaxel-platinum combination chemotherapy regimens used as the front-line postoperative adjuvant therapy are focused on the dose-dense paclitaxel, without consideration of platinum compounds or dose-dense both agents.

Study design:

It is not clear whether patients were randomized to each treatment and whether informed consent was obtained. The criteria on how each patient was assigned to a specific treatment is absent.

Response: Thank you very much for your valuable recommendation. Since both cisplatin and carboplatin are available in our department, administration of cisplatin or carboplatin was based on the patients’ will. We have demonstrated it in the text. Please read:

2.1. Patient Population

After institutional review board approval (VGHIRB 2019-07-039BC), all patients between January 2010 and December 2016 who fulfilled the following inclusion (International Federation of Gynecology and Obstetrics (FIGO) stage IIIC histologically confirmed high-grade serous-type ETOC and PPSC, an initial PCS, a total of six cycles of weekly paclitaxel [80 mg/m2] plus either triweekly cisplatin [20 mg/m2] or triweekly carboplatin [AUC 5] regimen) and exclusion (NACT, other newly diagnosed cancer, previous chemotherapy, or radiotherapy in the past two years; incomplete chemotherapy or the delayed of the first course chemotherapy (> 7 days after PCS), simultaneous use of other antineoplastic agents, antiangiogenic agents, or targeted therapy) criteria were identified from our gynecologic oncology registry. Since both cisplatin and carboplatin are available in our department, the selection of cisplatin or carboplatin was based on the patient’s will after shared-decision making done.  

Discussion subtitles:

  1. Main findings
  2. Comparison and discussion of finding A against the results of other studies and theoretical background
  3. The same for finding B, etc.

Response: It is of great value, and thank you very much. We have followed your recommendation to re-organize our finding and re-write the discussion. Please read:

4. Discussion

4.1. Main findings

The main findings of the current study showed that both regimens could be successfully used for the treatment of patients with FIGO IIIC ETOC or PPSC, offering 30 and 25 months of median PFS in paclitaxel-cisplatin and paclitaxel-carboplatin groups, respectively. Median OS was 58.5 months in paclitaxel-cisplatin group and 55.0 months in paclitaxel-carboplatin group. Although either PFS or OS did not reach the statistically significant difference, it seemed to be little better in patients treated with paclitaxel-cisplatin regimen compared to those treated with paclitaxel-carboplatin regimen. However, to compare the frequency and severity of AEs, we found that patients in paclitaxel-carboplatin group had more AEs than those in paclitaxel-cisplatin group did, including a higher risk of neutropenia and grade 3/4 neutropenia, and the need of the longer period to complete the front-line chemotherapy. The need of the longer period to finish the dose-dense therapy seemed to be associated with worse outcome of patients. Further analysis showed that the period between the first-time chemotherapy to the last dose (6 cycles) of chemotherapy > 21 weeks was associated with worse prognosis in patients compared to that ≤21 weeks, with hazard ratio (HR) of 81.24 for PFS and 9.57 for OS. As predicted, suboptimal debulking surgery (>1 cm) also contributed to the worse outcome than optimal debulking surgery (≤1 cm) with HR of 14.38 for PFS and 11.83 for OS.

4.2. Summary of Studies Addressing Dose-Dense therapy: Survival Outcome (Table 5)

Some early phase studies have also been conducted to test the tolerability and efficacy of dose-dense weekly paclitaxel (80 mg/m2) and triweekly carboplatin (AUC 5) with other agents [132,133]. For example, one phase II study modified a dose-dense paclitaxel (80 mg/m2) and every 4-week carboplatin (AUC 5) for the treatment of advanced-stage ETOC patients and found that median PFS and OS were 22.5 and 31.5 months, respectively [132]. Another phase II study with adding bevacizumab in dose-dense chemotherapy (weekly paclitaxel 80 mg/m2 and triweekly carboplatin AUC 5) showed the median PFS was 16.9-22.4 months in advanced stage ETOC patients [133].

Based on the impressive survival benefits of the earlier reports, the prospective, randomised trials may be the best method to test the efficacy and safety of dose-dense chemotherapy in the management of ETOC patients. The first large study, named as the JGOG 3016 has shown the survival benefits in dose-dense regimen containing weekly paclitaxel (80 mg/m2) and triweekly carboplatin (AUC 6), which not only offered a longer PFS (median PFS of 28.2 months, 95% CI 22.3-33.8 months) but also a significantly prolonged OS (median OS of 100.5 months, 95% CI 65.2-∞ months) than those in standard triweekly paclitaxel-carboplatin regimen [32,33].

The results of an NRG Oncology/GOG Study showed the similar PFS (the median PFS of 24.9 months) and OS (the median OS of 75.5 months) in dose-dense weekly paclitaxel (80 mg/m2) and triweekly carboplatin (AUC 6) with adding 15 mg/kg bevacizumab treatment compared to the standard triweekly combination therapy [72].

MITO-7 has shown that there was no statistically significant difference of median PFS between dose-dense weekly combination (paclitaxel 60 mg/m2 plus carboplatin AUC 2) and standard triweekly combination therapy (18.3 months [95% CI 16.8-20.9 months] versus 17.3 months [95% CI 15.2-20.2 months]) with HR of 0.96, 95% CI 0.80-1.16 [32].

GOG-0262 showed the median PFS of 14.9 and 14.2 months in dose-dense weekly paclitaxel (80 mg/m2) and triweekly carboplatin (AUC 6) with/without adding 15 mg/kg bevacizumab treatment, respectively, without statistically significant difference compared to the standard triweekly combination therapy [41].

The results from an ICON8 have shown that the median PFS was 20.8 months (95% CI 11.9-59.0 months) and 21.0 months (95% CI 12.0-54.0 months) in dose-dense weekly paclitaxel (80 mg/m2) and triweekly carboplatin (AUC 5 or 6) and in dose-dense weekly paclitaxel (80 mg/m2) and weekly carboplatin (AUC 2) regimen, respectively, and none of both was statistically significant different from the 17.7 months of the median PFS in standard-dose triweekly paclitaxel and triweekly carboplatin (AUC) treatment [33].

Although evidence might be much more powerful when the results were obtained from the prospective, randomised trials, the management of the individual patient who does not fulfill the criteria of a clinical trial or is enrolled into the clinical trial will be debated continuously. In addition, the strict criteria of a clinical trial, such as inclusive and exclusive criteria may not be reflective in real situations of patients. Therefore, clinical practices are not uniform, retrospective evaluations may partly reveal a real-world clinical practice. For example, elder population is often excluded in the clinical trials. Dr. Bun and colleagues found dose-dense weekly paclitaxel plus triweekly carboplatin was feasible for elderly patients although severe neuropathy might occur frequently compared with that in younger group [134]. In fact, elder population might have a higher risk to receive nonstandard chemotherapy [135].

Table 5. Summary of treatment efficacy and safety of dose-dense weekly paclitaxel plus platinum compounds in the management of patients with epithelial tubo-ovarian cancer and primary peritoneal serous carcinoma.

Authors

Stage

n

Regimen

PFS

OS

Prospective randomized trials

Katsumata [40]

II–IV

312

P 80 mg/m2 (D1,8,15),

C 6 (D1)

28.2 (M)

100.5 (M)

Pignata [32]

  IC-IV

406

P 60 mg/m2 (D1,8,15),

C 2 (D1,8,15)

18.3 (M)

Chan [41]

II–IV

340

P 80 mg/m2 (D1,8,15),

C 6 ±BEV (D1)

14.7 (M)

-

55

P 80mg/m2 (D1,8,15),

C 6 (D1)

14.2 (M)

-

Clamp [33]

I-IV

P 80 mg/m2 (D1,8,15),

C 5,6 ± BEV (D1)

20.8 (M)

P 80 mg/m2 (D1,8,15),

C 2 (D1,8,15) ± BEV (D1)

21.0 (M)

Walker [72]

II–IV

521

P 80 mg/m2 (D1,8,15),

C 6 +BEV (D1)

24.9 (M)

75.5 (M)

Retrospective study, including phase II study

Abaid [132]

III-IV

88

P 80 mg/m2 (D1,8,15),

C 5 (D1), stop one week

22.5 (M)

31.5 (M)

Fleming [133]

III-IV

33

P 80 mg/m2 (D1,8,15),

C 5 + BEV (D1)

16.9-22.4 (M)

Murphy [102]

III

38

P 80 mg/m2 (D1,8,15),

C 5 (D1)

31.3 (m)

54.5 (m)

Boraska Jelavić [105]

I-IV

43

P 80 mg/m2 (D1,8,15),

C 5 (D1)

20-24 (M)

Rettenmaier [78]

I-IV

100

P 80 mg/m2 (D1,8,15),

C 5 (D1)

27.6 (M)

Cheng [101]

IIIC-IV

32

P 80 mg/m2 (D1,8,15),

Cisplatin 20 mg/m2 (D1)

27.0 (M)

56 (m)

Current study

IIIC

18

P 80 mg/m2 (D1,8,15),

Cisplatin 20 mg/m2 (D1)

30.0 (M)

58.5 (M)

22

P 80 mg/m2 (D1,8,15),

C 5 (D1)

25.0 (M)

55.0 (M)

Dose-dense chemotherapy is limited on the intravenous route. RCT: randomized control trial, containing co-administration of bevacizumab (BEV) 15 mg/kilograms at day 1 triweekly and patients treated with neoadjuvant chemotherapy; n: number of patients; PFS: median (M) or mean (m) progression-free survival calculated by months; OS: median or mean overall survival calculated by months; Stage: FIGO stage (International Federation of Gynecology and Obstetrics stage); P: paclitaxel; D: day; C 2, C 5 or C 6: carboplatin (AUC 2, AUC 5 or AUC 6: area under the curve 2, 5, or 6 mg/mL per minute).

4.3. Dose-Dense Therapy-Related Adverse Events: Prefer the Use of Low-Dose Cisplatin in Place of Carboplatin in Platinum-based Therapy

It is believed in long-term that the use of carboplatin in place of original cisplatin in the platinum compound-based paclitaxel doublet might provide a better quality of life (QOL), since this regimen, with carboplatin dosed using the Calvert formula, yielded convincing non inferior outcomes when compared with the prior, more toxic, cisplatin-paclitaxel regimen [136]. By contrast, bone marrow suppression, including toxicity of hematological system might be more apparent in carboplatin administration, although Boyd and Muggia proposed that carboplatin-paclitaxel therapy is generally safe, when this drug is properly dosed [136]. Occurrence of neutropenia is one of the most causes to postpone the therapy or modify (decrease) the dosage of therapeutic agent. One study from Japan in 2006 tried to determine the feasibility of docetaxel-cisplatin therapy Q3W compared with docetaxel-carboplatin therapy Q3W as front-line for ETOC patients [136]. The results, as predicted, showed that incidence of grade 4 neutropenia was much more higher in the docetaxel-carboplatin group than that in the docetaxel-cisplatin group (74% versus 39%), suggesting the feasibility of docetaxel-cisplatin combination therapy as front-line therapy for ETOC patients [136]. In addition, rationale of the use “dose-dense” in place of “standard-dose” attempts to decrease the incidence of chemotherapy-related severe hematotoxicity, such as infection or sepsis caused by severe neutropenia or spontaneous hemorrhage caused by severe thrombocytopenia [49]. In Taiwan, GCSF is not prescribed in routine and insurance does not cover the cost if GCSF is used as prophylactic role. Furthermore, the obstacle is still present in facing patients with chemotherapy-related thrombocytopenia [137-139].

Thrombocytopenia is principal consideration and the dose-limiting toxicity of carboplatin [140,141]. Although thrombocytopenia unlikely occurs in a chemotherapy-naïve patients, but this toxicity usually begins to appear after day 14 and is predictably cumulative, contributing to the consideration of a carboplatin dose reduction [140]. Due to aforementioned limitation, some gynecological oncologists favored the use of cisplatin as the drug in the platinum-based combination therapy, except for those patients with identified impaired renal function (eGFR < 60 ml/min). The current low-dose cisplatin plus dose-dense paclitaxel regimen was associated with lower rates of hematologic toxicities compared to the conventional dose-dense paclitaxel plus carboplatin regimen. The results from an ICON8 have also shown that patients treated with weekly regimen had increased grade 3 or 4 toxic effects, although these high-grade toxicities were predominantly uncomplicated [33]. However, both regimens are carboplatin-based paclitaxel combination.

Consistent with more toxicity of the use of carboplatin in previous study [136], the incidence of grade 3/4 neutropenia was significantly lower in the cisplatin-paclitaxel arm than that in the carboplatin-paclitaxel arm (27.8% versus 77.3%, p = 0.002) in our study. In addition, other parameters of hematological examination, including grade 3/4 anemia and grade 3/4 thrombocytopenia also favored the cisplatin-paclitaxel regimen with incidence of 5.6% and none, respectively compared to 22.7% and 13.6% in the carboplatin-paclitaxel regimen in the current study. We believed that this carboplatin-paclitaxel combination therapy-related hematotoxicity might explain more patients in the carboplatin-paclitaxel arm had a prolongation of the period between the initial first time of chemotherapy and the end of final chemotherapy (six cycles of chemotherapy) than those in the cisplatin-paclitaxel arm did (45% versus 11.1%, p = 0.018). Further survival analysis showed this prolongation of therapeutic period might be associated with bad outcome of FIGO IIIC ETOC and PPSC patients.

By contrast, the cisplatin-related renal, neural, and gastrointestinal system toxicity was lower in the current study. The main reason is the use of low-dose cisplatin (20 mg/m2) as a platinum compound in the platinum-paclitaxel combination therapy.        

4.4. The Benefits of Maximal Cytoreductive Surgery and The Consideration of the Location of Residual Tumors

In agreement with well-known concept [142-146], a rapid reduction of the tumor burden is the guarantee of the successful treatment in cancer patients. Nearly all studied have confirmed that maximal cytoreduction surgery is the most critical step in the management of patients with advanced-stage ETOC or PPSC [142-146]. According to the definition of the Gynecologic Oncology group (GOG) for 'optimal' as having residual tumor nodules each, which should be ≤ 1 cm in size, Cochrane review revealed that outcome of ETOC patients with residual disease ≤ 1 cm is better than that with residual disease > 1 cm [143], although it is well known that PCS to no gross residual tumor might be associated with the longest PFS and OS [144,145]. One study showed an interesting finding that patients still had diverse outcomes despite undergoing optimal PCS (≤ 1 cm residual disease), in which it demonstrated that patients with tumor limited in a single anatomic location had better median PFS and OS than patients with tumors widely distributed in multiple anatomic locations [146].

In our current study, we also tested the aforementioned findings. We found that patients with suboptimal PCS with a residual tumor > 1 cm indeed had a worse prognosis with a hazard ratio (HR) of 14.38 (95% confidence interval [CI] 4.18-49.46) in PFS and a HR of 11.83 (95% CI 1.48-94.72) in OS, comparing with those patients who could achieve optimal PCS with a residual tumor in size less than 1cm, supporting the rationale that small-volume tumors are more prone to clearance [61]. However, the widespread of tumor seemed to be not an independent risk factor for worse prognosis. We found that there was no statistically significant difference of outcome in patients whose residual tumors in localized anatomical site or in multiple anatomical sites.

One of the limitations of this study is small sample size, which resulted in wide confidence intervals demonstrating low precision of the estimates. This must be stated and the results should be qualified as indicative or suggestive.

Response: Thank you very much for your comments and it is a real and big weak point. However, we would like to comment this part.

4.5. The Strength and The Limitation

The current study has the following strength. All patients were serous type ETOC and PPSC with FIGO stage IIIC. The follow-up period was long enough with nearly 5 years (a mean follow-up period of 55 months). All patients were treated with by one senior doctor (P.-H.W). All suggested study population was homogeneous.

There are some limitations in the current study. Retrospective data collection in nature may not be strong to show the evidence. However, as shown above, it is a real-world clinical practice. In addition, only patients who underwent PCS with surgical confirmation of FIGO IIIC serous-type ETOC and PPSC and complete 6 cycles of dose-dense paclitaxel-platinum compounds were included, contributing to the risk of selection bias. Furthermore, number in the current study was small in each arm, contributing to wide confidence intervals. It demonstrated the higher risk of low precision of the estimates. Therefore, the findings of the current study should be used in caution. Finally, new therapeutic strategies, such as maintenance therapy or NACT or intraperitoneal therapy were not evaluated in the current report. However, we believe none of them will impede the value of current study.

5. Conclusions

Our study found that paclitaxel might be one of the most critical components for the successful administration of dose-dense adjuvant chemotherapy. In addition, on-schedule and shortening of the entire therapeutic period are a critical step for the survival benefits of patients who underwent a postoperative adjuvant dose-dense paclitaxel plus platinum compounds regimen. Of course, maximal efforts should be done to eradicate the tumor as much as possible. Although recent evidences highlighted the value of precise medicine, personalized and individual therapy along with targeted therapy for cancer patients [147-151], the net health benefits (NHBs), cost-effectiveness or quality of life may reflect the relative value of treatment options in EOC [100, 152-157]. With low risk of AEs, and shortening therapeutic period, and of most importance, without compromising therapeutic effects, low-dose triweekly cisplatin plus dose-dense weekly paclitaxel might be associated with the better NHBs, more cost-effectiveness and a better QOL compared to carboplatin-based dose-dense paclitaxel regimen. More studies are welcome to test our findings.

Overall, this manuscript will benefit from better more clear organization of the material.  

Response: Thank you very much for your comment. We have tried our best to re-organize the text and your favorable consideration is much appreciated.
